# MisAttributionLLM: Integrating Error Attribution Capability into LLM Evaluation

## Abstract

With the widespread application of Large Language Models (LLMs) in various tasks, evaluating the performance of LLMs becomes an essential research topic. However, existing judge models lack the specific capability required for error attribution (i.e., identify the types of error made in responses). In this work, we first establish a comprehensive Misattribution Framework with 9 primary and 19 secondary categories, which are intended to facilitate in-depth analysis and enhance the performance of LLMs. Based on this framework, we present AttriData, a dataset specifically designed for error attribution, encompassing misattributions, along with the corresponding scores and feedback. We also propose MisAttributionLLM, a fine-tuned model on AttriData, which is the first open-source, general-purpose judge model with error attribution capability which provides valuable insights into the model's weaknesses and enables targeted improvements. Experimental results show that MisAttributionLLM achieves the highest Pearson correlation with human evaluators among 8 open-source and closed-source LLMs. Furthermore, MisAttributionLLM also obtains the highest accuracy and micro-F1 in the performance of error attribution. Extensive experiments and analyses are conducted to confirm the effectiveness and robustness of our proposed method. [1]

## 1 Introduction

With the rapid development of large language models(LLMs), assessing the performance of LLMs has become a vital research topic (Xie et al., 2023; Chang et al., 2024; Liu et al., 2024). A solid evaluation method is capable of providing high-quality opinions to guide the LLM in its continuous improvement (Kim et al., 2023b).

The application of LLM-as-a-Judge model (Liu et al., 2023; Zheng et al., 2024) has drawn significant attention because of its potential to rival human assessment. Access to high-performing large language models such as GPT-4 (Achiam et al., 2023) is generally limited to the OpenAI API due to their proprietary. Considering the need to avoid potential risks of commercial APIs like high cost, unstable usage, and data leakage, researchers have commenced training their own judge models (Kim et al., 2023b; Ke et al., 2024; Wang et al., 2023b). For instance, Kim et al. (2023b) proposes PROMETHEUS, an open-source language model designed to induce fine-grained evaluation with feedback, which provides a detailed explanation for why a given answer would be awarded a specific score. Nevertheless, both the above open-source and closed-source LLMs focus on scores and feedback, which is insufficient for in-depth analysis and model's targeted improvements.

Specifically, when evaluating the performance of a large language model, it is essential to focus on the instances where the model answers display undesirable and inconsistent behaviors, commonly referred to as error responses (Kamoi et al., 2024b; Chen et al., 2023). Error responses encompass a variety of issues, such as generating outputs that are convincingly presented but factually incorrect or misleading, known as hallucinations (Lin et al., 2022; Zhang et al., 2023a), engaging in reasoning that does not align with the established facts or context, termed unfaithful reasoning (Golovneva et al., 2023; Lyu et al., 2023), and failing to adhere to specified rules or constraints (Zhuo et al., 2023; Wang et al., 2023a). These behaviors undermine the confidence in LLMs and pose substantial challenges to their practical deployment. Unfortunately, researchers concentrate solely on score and

---

[1]The codes, datasets and models will be released after the peer review.

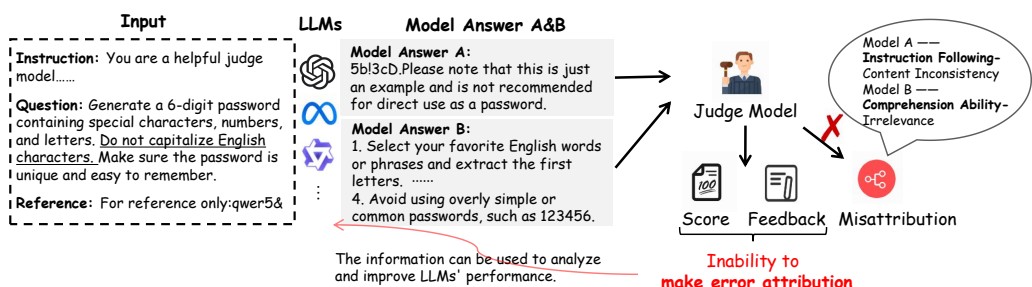

Figure 1: The existing judge model is capable of outputting score and feedback, but it lacks the ability to make error attribution. The capability of error attribution is helpful in analyzing and improving the performance of LLMs.

feedback during evaluation and neglect to identify the types of error made in these error responses, as depicted in Figure 1. This oversight tends to result in a misjudgment of the LLM's performance and is likely to hinder the identification of critical opportunities for improvement. Therefore, the systematic classification of error responses, known as error attribution, is a critical aspect of LLMs' analysis and is vital for enhancing their robustness and effectiveness.

To this end, we propose MisAttributionLLM, a 7B LLM with error attribution capability that is not only equipped to score the LLMs' responses and generate appropriate feedback but also able to provide detailed misattributions. We first establish a comprehensive Misattribution Framework with 9 categories at the first level and 19 at the second level, to facilitate subsequent analysis and improvement of LLMs. Based on Misattribution Framework, we present AttriData, a high-quality Chinese dataset that is manually annotated and crafted to encompass a variety of comprehensive evaluation tasks, representing realistic user demands. The dataset includes approximately 20,000 samples. With Misattribution Framework, we aim to set a new standard and benchmark in the evaluation of LLMs.

By fine-tuning Qwen2-7B (Yang et al., 2024) with the AttriData, we obtain the MisAttributionLLM. On the AttriData test dataset, MisAttributionLLM achieves a Pearson correlation of 0.931 with human evaluators, which is higher than GPT-4 (0.802), and significantly exceeds GPT-3.5 (0.410). In terms of the performance of error attribution, MisAttributionLLM achieves a micro-F1 score of 0.813, surpassing 7 open-source and closed-source LLMs. Furthermore, when human evaluators are tasked with selecting the higher-quality feedback in pairwise comparisons, MisAttributionLLM is chosen over GPT-4 in 53.68% of the cases and outperforms GPT-3.5 with an 83.82% win rate. To the best of our knowledge, we are the first to propose a Misattribution Framework that is capable of being applied for systematic model answer analysis, which leads to insightful evaluation and improvement of LLMs.

In conclusion, our work delivers three key contributions:

- We establish a comprehensive Misattribution Framework, which consists of 9 primary and 19 secondary categories, to facilitate the subsequent analysis and enhancement of LLMs. Utilizing this framework, we introduce AttriData, specifically developed for incorporating error attribution capability into LLM evaluation. Unlike previous datasets, AttriData includes misattributions in addition to scores and feedback.

- We propose MisAttributionLLM, the first open-source, general-purpose large language model capable of error attribution and specifically designed for fine-grained evaluation. This innovation provides valuable insights into the model's potential shortcomings, enabling targeted adjustments and improvements to enhance its performance.

- We conduct extensive experiments demonstrating the effectiveness of incorporating error attribution capability into LLM evaluation. MisAttributionLLM shows a strong correlation with human evaluators in scoring setting, and achieves high accuracy and micro-F1 in the performance of error attribution. Extensive experiments and analyses are conducted to confirm the effectiveness and robustness of our proposed method.

## 2    RELATED WORK

**Evaluation Method**    With the development of large language models (LLMs), recent studies have employed GPT-4 or fine-tuned LLMs as judge models (Kim et al., 2023b; Jiang et al., 2023; Wang et al., 2023b; Ye et al., 2024). For example, Wang et al. (2023b) introduces PandaLM, a fine-tuned LLM designed to assess generated text and provide explanations regarding its reliability across various preference datasets. PROMETHEUS (Kim et al., 2023b) stands out as an open-source LLM tailored for fine-grained evaluation, capable of adapting to a wide range of scoring rubrics. Moreover, CritiqueLLM (Ke et al., 2024) demonstrates the beneficial effects of generated critiques as scalable feedback, enhancing the quality of LLM outputs. TIGERScore(Jiang et al., 2023) is guided by natural language instruction to provide error analysis to pinpoint the mistakes in the generated text. More evaluation methods are detailed in Appendix A.

**Error Attribution**    Although the error attribution has not been formally proposed, research related to it has primarily been conducted in the context of enhancing LLM responses using feedback from LLMs (Kamoi et al., 2024b; Chen et al., 2023; Gou et al., 2024; Pan et al., 2024). These studies focus on self-correction during training but often neglect the analysis of model responses with misattribution. Kamoi et al. (2024a) have begun to address the issue of errors in model responses, but the types of error they identify are limited, and there is a lack of trained judge models to tackle these issues. To address these limitations, we propose a comprehensive Misattribution Framework and train the judge model to be capable of error attribution.

## 3    METHOD

An overview of our method is illustrated in Figure 2. The process can be generally divided into three main steps: data construction, supervised fine-tuning, and inference. The Misattribution Framework is described in detail in Section 3.1. In Sections 3.2 and 3.3, we present the construction and analysis of AttriData. Lastly, the fine-tuning procedure for MisAttributionLLM is outlined in Section 3.4.

### 3.1    MISATTRIBUTION FRAMEWORK

We conduct a thorough and in-depth analysis of the error responses associated with LLMs (Zhang et al., 2023a; Lyu et al., 2023; Wang et al., 2023a; Kamoi et al., 2024b) and propose a detailed and systematic Misattribution Framework. This framework consists of 9 primary categories and 19 secondary categories, effectively capturing the current limitations of LLMs across various application scenarios. The categories are illustrated in Figure 12. The primary categories encompass critical dimensions such as Response Quality (Yin & Wan, 2022), Instruction Following (Zeng et al., 2024), Knowledge Ability (Ji et al., 2023; Zhang et al., 2023b; Pagnoni et al., 2021), Reasoning Capability (Bhargava & Ng, 2022), Comprehension Ability (Shi et al., 2023), Creative Ability (Ismayilzada et al., 2024), Safety (Qiu et al., 2023), Multi-Turn Dialogue (Yi et al., 2024) and Other Errors. Detailed explanations of these categories are provided in Table 1 and the detailed examples can be referenced in Figure 9, 10 and 11.

### 3.2    DATASET CONSTRUCTION

**Data Collection**    We cooperated with prominent data companies to obtain the critical issues from the current applications of LLMs, such as ERNIE Bot[2] and Hunyuan[3]. Inspired by Xie et al. (2023), we manually labeled the data based on TencentLLMEval's task tree. These issues can be categorized using a seven-level classification system: NLP Basic, Multi-Turn Dialogue, Math, Reasoning, Text Generation, Question and Answer, Professional Field. Our selection of data is guided by two primary considerations:

- **Comprehensive Evaluation Tasks:** The evaluation tasks included in this dataset are designed to address both fundamental and advanced performance of LLMs comprehensively.

---

[2]https://yiyan.baidu.com
[3]https://hunyuan.tencent.com

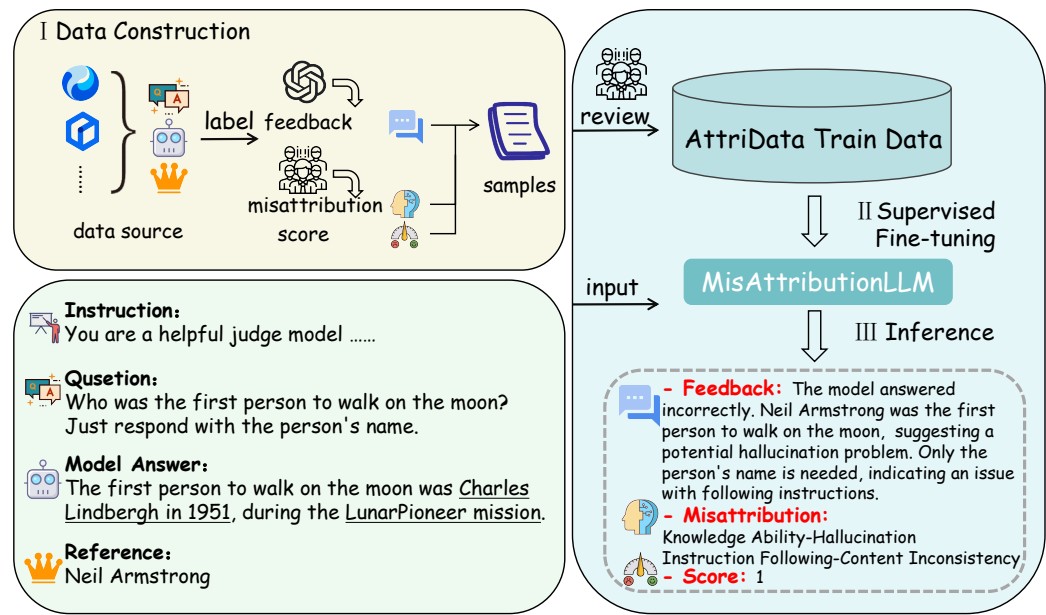

Figure 2: The overview of our method. The process can be generally divided into three main steps: data construction, supervised fine-tuning, and inference. The Data Construction of AttriData's annotations includes score, misattribution, and feedback. After annotations, we collected the AttriData which we fine-tuned on Qwen2. A sample of the dataset is visually represented by a rectangular box with a green background. This sample is input into the MisattributionLLM model, and the resulting inference is displayed within a gray dashed box.

- **User-Driven Focus:** The data focuses on issues that are of significant concern to users. Originating from real-world application scenarios, it provides an accurate reflection of the current public demand for LLMs.

The entire data collection process lasted three months, and resulted in the collection of 22,832 data. Details of the data are provided in Table 6.

**Annotation Workflow**   To ensure consistency and accuracy in the annotation process, the annotators should first familiarize themselves with the specific guidelines for each section of the dataset: score, misattribution, and feedback. As illustrated in Figure 2 (I), the annotators' task is to read the question, reference answer, and model answer, and then annotate accordingly. Firstly, scores range from 0 to 3 points: 3 point is awarded if the model provides a correct answer without any errors; 2 point is given if the answer is partially correct; 1 point is assigned if the answer is completely incorrect; and 0 point is given if the model provides an off-topic response or violates safety guidelines. Inspired by the score setting of Lin et al. (2024), we have chosen this distribution of scores, which provides a clear and precise representation of the quality of responses corresponding to each score.

For misattribution, if the score is less than 3 point, the annotators need to identify the types of error in the model answer referring to Misattribution Framework. The specific Misattribution Framework is described in Section 3.1.If the score is 3 point, the misattribution is marked as NULL. For feedback, we are inspired by (Kim et al., 2023b) and use GPT-4 to generate the feedback. The generated template is shown in Figure 6.

We organized 36 annotators and 12 senior annotation experts [4], all of whom were thoroughly trained in the annotation guidelines. To ensure quality, each data is independently annotated by three annotators and subsequently reviewed by one senior expert. In cases where the three annotators produce inconsistent results, a senior expert conducts a careful review to identify any potential errors or omissions and makes the final determination. In addition, we divided the data into 20 batches and

---

[4]The evaluation cost is $1,000 per person per month.

Table 1: The overview of Misattribution Framework. Explanations for each second-level category under the first-level categories.

| First-level category | Second-level category | Explanation |
|---|---|---|
| Response Quality (Yin & Wan, 2022) | Typos | The response includes spelling errors. |
| | Noisy | The response includes irrelevant or redundant information. |
| | Truncation | The model's response is cut short, resulting in an incomplete answer. |
| | Duplicate | The response contains repeated information. |
| | Refusal to Answer | The model refuses to provide an answer. |
| | Missing Answers | Multiple questions are asked, but responses are provided for only a portion of them. |
| Instruction Following (Zeng et al., 2024) | Content Inconsistency | The text generated by the model fails to meet the required content standards, such as language, structure, theme and style. |
| | Format Inconsistency | The response does not conform to the constraints specified in the instructions. |
| | Length Inconsistency | The length of the response does not align with the requirements outlined in the instruction, such as word count, number of paragraphs, or number of sentences. |
| Knowledge Ability (Ji et al., 2023) | Hallucination | It refers to the phenomenon in which the content generated by the model is inconsistent with real-world facts or the user's input. |
| | Incorrect Answers | This primarily refers to objective questions where the response does not match the correct answer. |
| Reasoning Capability (Bhargava & Ng, 2022) | Process Error | This occurs when there are logical flaws in the reasoning process. |
| | Result Error | Errors in the final outcomes of reasoning, particularly in disciplines like mathematics and coding. |
| Multi-Turn Dialogue (Yi et al., 2024) | Reference Error | There is a failure to understand the content that reference pronouns refer to. |
| | Long-term Memory Loss | The inability to incorporate contextual information into responses. |
| Creative Ability (Ismayilzada et al., 2024) | Inappropriate Content | The content generated by the model fails to align with the creative requirements. |
| Safety (Qiu et al., 2023) | Safety Concerns | This type of error involves the potential harm posed to users or society. |
| Comprehension Ability (Shi et al., 2023) | Irrelevance | The response provided does not adequately address or answer the specific question that was asked. |
| Other Errors | Others | This category encompasses errors that do not fit into the aforementioned classifications. |

randomly selected 30% of the submissions from the senior annotation experts for quality checks. If the accuracy of these checks falls below 98%, the corresponding batch is sent back for re-annotation. Overall, the entire annotation process took approximately three months to complete.

### 3.3 DATASET ANALYSIS

**Dataset Statistics**  The dataset consists of 22,832 samples, of which 14,295 are satisfactory without misattribution, and 8,537 with misattribution, 2,031 of which are multi-label. The vast majority of the AttriData is in Chinese, with an additional 1,321 data in English. The training set contains 19,835 samples, while the testing set contains 2,997 samples. Detailed statistics are presented in Table 2. Unlike previous datasets, AttriData is distinguished by the inclusion of samples with misattribution, a feature that has not been addressed before. The information about the amount of misattribution data is shown in Table 7. Furthermore, the labels of score and misattribution in AttriData have been manually annotated, different from other benchmarks, which are generated by LLMs. Annotation generated by LLMs is susceptible to limitations and unreliability due to the inherent constraints of the models themselves, whereas our manual annotation offers a substantial advantage in terms of reliability.

**Dataset Quality**  Given that the batch annotation method we developed ensures a certain degree of annotation accuracy, we further assess the level of agreement among multiple annotators. Specifically, we compute Fleiss' kappa (Moons & Vandervieren, 2023) to evaluate the consistency in labeling the scores and misattributions of the data. The resulting kappa values are 0.875 and 0.832, respectively, suggesting that our annotations can be regarded as almost perfect agreement (Landis, 1977).

### 3.4 FINE-TUNING LANGUAGE MODEL

We utilize AttriData to fine-tune Qwen2-7B and obtain MisAttributionLLM, equipping it with the capability of error attribution. Following the approach of Chain-of-Thought Fine-Tuning(Kim et al., 2023a; Yao et al., 2024), our fine-tuning process involves sequentially generating feedback, identifying misattribution, and then assigning a score. Figure 2 (II) illustrates the supervised fine-tuning process. Utilizing the AttriData training dataset, we fine-tuned Qwen2 to attain the MisAttribution-LLM. For inference, as depicted in Figure 2 (III), given a instruction, a question, a model answer text, and a reference text, the objective is to produce a comprehensive result including a rating score, a misattribution, and feedback. The detailed prompt utilized can be found in Figure 7 and 8 for English and Chinese respectively. For all LLMs, we use uniform prompt. The details of fine-tuning and inference procedures are provided in Section 4.2.

## 4 EXPERIMENTS

In this section, we explain our experiment setting, which includes the list of experiments, metrics, and baselines that we used to evaluate the performance of LLMs.

### 4.1 BASELINES

The following lists outline the baselines we employed for comparison in experiments. They include both open-source and closed-source large language models:

- **Llama3.1-8B (Dubey et al., 2024):** is a top-performing open-source language model, known for its adaptability across various natural language processing tasks.

- **Qwen2-7B (Yang et al., 2024):** serves as the base model for MisAttributionLLM and is a leading choice among open-source models for Chinese language processing, also acting as an evaluator in this study.

- **GLM4-9B (GLM et al., 2024):** stands out as an exceptional open-source large language model optimized for Chinese language tasks.

- **GPT-3.5-turbo-0613(GPT-3.5) (Ouyang et al., 2022):** is a closed-source large language model offering a cost-effective alternative for evaluation purposes.

- **GPT-4-1106-preview(GPT-4) (Achiam et al., 2023):** is recognized as one of the most robust closed-source models, often chosen as the primary judge model in language model evaluation.

- **ERNIE-4.0-8K (Tang et al., 2024):** is a leading closed-source model for Chinese large language processing, noted for its advanced natural language understanding and generation capabilities.

- **Doubao-pro-4K (Doubao Team, 2024):** is a widely adopted Chinese large language model, popular for its applications in diverse real-world scenarios.

### 4.2 IMPLEMENTATION DETAILS

We choose Qwen2-7B (Yang et al., 2024) as our base model and implement Zero Redundancy Optimizer (ZeRO) (Rajbhandari et al., 2020) stage 3 framework from the Deepspeed library (Rasley et al., 2020). MisAttributionLLM is trained on 8 A100 GPUs. We employ the AdamW optimizer (Kingma, 2015) with the weight decay of 0.1. The learning rate is set at 1.0e-4, accompanied by a warmup ratio of 10%. The batch size is set to 16 and the number of training epochs is 3. We utilize a training set consisting of 19,835 samples and a testing set comprising 2,997 samples from the AttriData dataset. The details of the AttriData dataset can be found in Table 2. We conduct experiments in which the judge models generate feedback, misattribution, and score based on the provided instruction, question, model answer, criteria, and reference. By integrating these components, our method aims to offer a comprehensive evaluation of the model's performance.

Table 2: Statistics of datasets. Comparison between AttriData and existing benchmark. GPT-4 assisted indicates that the feedback was generated with the help of GPT-4, whereas human assisted means that humans were involved in reviewing the scoring.

| Dataset | Sum | Train | Test | Misattribution | Train Annotation | Test Annotation |
|---------|-----|-------|------|----------------|------------------|-----------------|
| PROMETHEUS | 21,000 | 21,000 | 1,000 | ✗ | GPT-4 | GPT-4 |
| CritiqueLLM | 36,815 | 35,815 | 1,000 | ✗ | ChatGPT | GPT-4(human assisted) |
| **AttriData** | 22,832 | 19,835 | 2,997 | 8,537 | Human(GPT-4 assisted) | Human(GPT-4 assisted) |

Table 3: Pearson, Kendall-Tau, Spearman correlation coefficients on AttriData test dataset. The best comparable statistics are **bolded** and second best underlined.

| Evaluator LM | AttriData-Test | | |
|--------------|----------------|----------|-------------|
| | **Pearson** | **Spearman** | **Kendall-Tau** |
| Llama3.1-8B | 0.482 | 0.476 | 0.451 |
| Qwen2-7B | 0.154 | 0.173 | 0.161 |
| GLM4-9B | 0.620 | 0.609 | 0.571 |
| Doubao-pro-4K | 0.672 | 0.681 | 0.644 |
| ERNIE-4.0-8K | 0.798 | 0.827 | 0.781 |
| GPT-3.5 | 0.410 | 0.406 | 0.381 |
| GPT-4 | 0.802 | 0.832 | 0.786 |
| MisAttributionLLM-7B | **0.931** | **0.942** | **0.927** |

### 4.3 MAIN RESULTS

**Correlation with Human Scoring**  Following (Ke et al., 2024), we utilize Pearson, Spearman, and Kendall correlation coefficients to evaluate the performance of the judge models. The detailed metrics can be found in Appendix B. Specifically, these coefficients measure the agreement between human judgments and evaluation scores across all generated samples for each instruction from the judge models. The correlation values are calculated based on the scores derived from these coefficients.

The results, which are presented in Table 3, indicate that among all the models evaluated, MisAttributionLLM achieves the highest scores on all three correlation coefficients, outperforming both open-source and closed-source LLMs. This highlights the superiority and effectiveness of MisAttributionLLM in scoring setting task. Among the closed-source models, GPT-4 is a close second, while ERNIE-4.0-8K also shows commendable performance. However, the open-source model Qwen2-7B exhibits relatively lower results, which underscores the critical role of fine-tuning based on AttriData. The performance of Llama 3.1-8B is also unsatisfactory, as it is primarily optimized for English and performs poorly in Chinese.

**The Performance of Error Attribution**  To evaluate the performance of error attribution in LLMs, we measure from two perspectives: the detection of misattribution and the multi-classification of misattribution. For misattribution detection, this refers to whether the judge model correctly determines that there is an error in the model response. For multi-classification of misattribution, this refers to the process of error attribution (i.e. whether the error is correctly categorized). For misattribution detection, we adopt precision, recall, and F1 score to assess the performance of the judge models. For multi-classification of misattribution, we use accuracy and micro-F1 score (Harbecke et al., 2022) to evaluate the capability of the judge models.

The results of the error detection and the multi-classification of misattribution are detailed in Table 4. MisAttributionLLM demonstrates exceptional performance in error detection, achieving the highest accuracy of 98.1%, which signifies its high proficiency in identifying errors. In terms of recall, GPT-4 excels, suggesting a propensity for error attribution. This characteristic could be advantageous in contexts where the cost of failing to identify an error outweighs that of incorrectly flagging an error.

Regarding the multi-classification of misattribution, MisAttributionLLM surpasses other LLMs, not only in terms of accuracy but also in micro-F1 score, outperforming its closest competitors by a

Table 4: The results of the misattribution detection and the multi-classification of misattribution on AttriData test dataset. The best comparable statistics are **bolded** and second best underlined.

| Evaluator LM | Misattribution Detection | | | Multi-Classification | |
|---|---|---|---|---|---|
| | Precision | Recall | F1 | Accuracy | Micro-F1 |
| Llama3.1-8B | 0.485 | 0.422 | 0.478 | 0.463 | 0.442 |
| Qwen2-7B | 0.572 | 0.491 | 0.541 | 0.570 | 0.484 |
| GLM4-9B | 0.538 | 0.888 | 0.670 | 0.526 | 0.574 |
| Doubao-pro-4K | 0.759 | 0.895 | 0.821 | 0.648 | 0.674 |
| ERNIE-4.0-8K | 0.802 | 0.946 | 0.868 | 0.700 | 0.721 |
| GPT-3.5 | 0.486 | 0.772 | 0.725 | 0.482 | 0.542 |
| GPT-4 | 0.814 | **0.955** | 0.879 | 0.708 | 0.715 |
| MisAttributionLLM-7B | **0.981** | 0.954 | **0.967** | **0.820** | **0.813** |

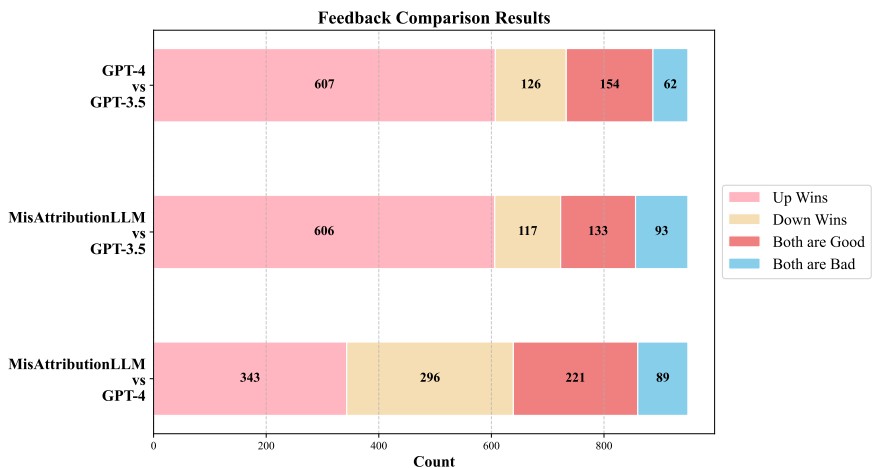

Figure 3: The results of pairwise comparison of the quality of the feedback generated by GPT-4, MisAttributionLLM and GPT-3.5. Annotators are asked to select which feedback is better at evaluating the given response. MisAttributionLLM shows a win-rate of 53.68% over GPT-4 and 83.82% over GPT-3.5.

significant margin of over 9%. The major advantage underscores the robustness and effectiveness of MisAttributionLLM in handling complex multi-classification task. Overall, the results indicate that closed-source LLMs generally outperform open-source LLMs. However, the cost and lack of transparency of closed-source LLMs may limit their adoption. In contrast, our fine-tuned open-source model, MisAttributionLLM, consistently matches or exceeds the performance of other LLMs, which can rival or surpass closed-source solutions in specialized tasks.

**Pairwise Comparison of the Feedback with Human Evaluation** To assess the quality of the generated feedback, we conduct pairwise comparisons among the feedback produced by MisAttributionLLM, GPT-3.5, and GPT-4. Human evaluators are tasked with selecting which feedback they believe is of higher quality at the aspect of score and misattribution (i.e., win, lose, or tie) and providing their reasoning for this choice. We select 949 samples from AttriData test dataset for pairwise comparisons. Specifically, we recruit 9 annotators and divide them into three groups: one group comparing MisAttributionLLM with GPT-4, another comparing MisAttributionLLM with GPT-3.5, and the last group comparing GPT-4 with GPT-3.5. The source of the feedback is anonymous to the annotators. The results are shown in Figure 3, demonstrating that MisAttributionLLM is preferred over GPT-4 53.68% of the times and over GPT-3.5 83.82% of the times. Since the feedback is generated by GPT-4, GPT-4 performs relatively well. These findings indicate that the feedback provided by MisAttributionLLM is not only meaningful and insightful but also highly beneficial for improving the accuracy of scoring and error attribution.

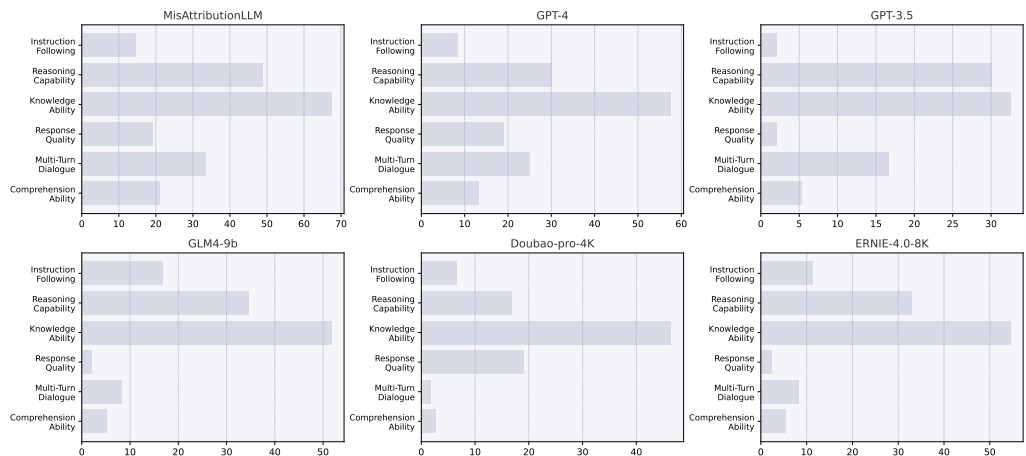

Figure 4: The results of the error attribution capability of six LLMs across six aspects.

## 5 DISCUSSIONS AND ANALYSIS

### 5.1 ABLATION STUDY

To further investigate the impact of misattribution on MisAttributionLLM in scoring setting, we utilize Pearson, Spearman, and Kendall correlation coefficients to evaluate the performance of MisAttributionLLM. We remove the misattribution information from the AttriData because error attributon is a more complex capability. The results presented in Table 5 indicate that the performance of MisAttributionLLM has a minor decrease but is not significantly affected in the absence of misattribution. The results are consistent with our hypothesis that misattribution plays an auxiliary role in evaluating the model in scoring setting.

### 5.2 ANALYSIS OF MODEL ERROR ATTRIBUTION CAPABILITIES IN LLMS

To explore the model's ability to distinguish samples with different types of misattribution, we conduct a detailed analysis of six LLMs (MisAttributionLLM, GPT-4, GPT-3.5, GLM4-9B, Doubao-pro-4K, and ERNIE-4.0-8K) across six aspects: response quality, multi-round dialogue, reasoning ability, knowledge ability, comprehension ability, and instruction following, with a focus on the accuracy for samples with misattribution. The results are illustrated in Figure 4. Notably, MisAttributionLLM performs well in knowledge ability and reasoning capability, but it falls short in multi-turn dialogue and instruction following. The discrepancy likely arises from the insufficient training data for these two aspects, suggesting a potential area for future enhancement. In conclusion, the performance of the six LLMs varies significantly across the evaluated skills, providing an objective reflection of their capabilities in various dimensions.

Table 5: Pearson, Kendall-Tau, Spearman correlation coefficients on AttriData test dataset.

| Evaluator LM | AttriData-Test | | |
|---|---|---|---|
| | **Pearson** | **Spearman** | **Kendall-Tau** |
| MisAttributionLLM-7B | 0.931 | 0.942 | 0.927 |
| w/o Misattribution | 0.921 | 0.939 | 0.925 |

### 5.3 CASE STUDY

In Figure 5, we present several cases to illustrate that, after fine-tuning on the AttriData, MisAttributionLLM is capable of generating feedback, misattribution, and score. These capabilities that are comparable to, and sometimes exceed those of, GPT-4. In the first case, GPT-3.5 appears to have misunderstood the input, leading to a response that contradicts the fundamental fact that binary

numbers consist solely of 0s and 1s. Both GPT-4 and MisAttributionLLM accurately detect the error in the original model's response and correctly classify it as an issue related to knowledge ability, specifically the hallucination. In the second case, although GPT-4 provides correct feedback and score, it is not as accurate in terms of misattribution compared to MisAttributionLLM. The superior performance of MisAttributionLLM in error attribution capability reflects a deeper understanding of the nuances of task contexts and a more accurate evaluation of the model's answers.

| Input & Gound Truth | Responses from different LLMs | Results |
|---|---|---|
| **Question:** The first and last digits of an 8-bit binary number are both 2. Is it divisible by 2 after conversion to decimal?

**Model Answer:** The lowest bit of the binary number is 2, which means it is actually an even number. Therefore, no matter what the other bits of this binary number are, it is divisible by 2 in decimal.

**Reference:** There is no digit 2 in binary numbers, so it is impossible to determine whether it is divisible by 2.

**Groud Truth:**
- Feedback: The model answer is wrong and there are logical traps. There is an hallucination problem in that 2 does not actually exist in binary digits.
- Misattribution: Knowledge Ability-Hallucination
- Score: 1 | **GPT-4:**
- Feedback: The model's reasoning is wrong. There is no 2 in binary numbers, binary numbers only contain 0 and 1.
- Misattribution: Knowledge Ability-Hallucination
- Score: 1

**GPT-3.5:**
- Feedback: The model answered with the correct binary number conversion and reached the correct conclusion.
- Misattribution: NULL
- Score: 3

**MisAttributionLLM:**
- Feedback: The model answered incorrectly. The number 2 does not exist in binary and the model made an error in understanding the question.
- Misattribution: Knowledge Ability-Hallucination
- Score: 1 | ✓


✗



✓ |
| **Question:**
Generate 8-digit passwords containing numbers and letters that are not repetitive and easy to remember.

**Model Answer:**
1. Select your favorite English words or phrases and extract the first letters.
2. Use your favorite English words, pinyin or abbreviations, plus numbers or special symbols.

**Reference:** For reference only:643chk1d

**Groud Truth:**
- Feedback: The model did not generate a 8-digit password, but instead provided suggestions for generating a password. The model's comprehension ability was problematic and the answer was irrelevant to the question.
- Misattribution: Comprehension Ability-Irrelevance
- Score: 1 | **GPT-4:**
- Feedback: The model fails to generate specific password instances as instructed, but instead provides general suggestions for creating passwords.
- Misattribution: Instruction Following-Format Inconsistency
- Score: 1

**GPT-3.5:**
- Feedback: There were content inconsistency errors in the model responses.
- Misattribution: Response Quality-Format Inconsistency
- Score: 2

**MisAttributionLLM:**
- Feedback: The model answer did not directly generate a password, but gave a general method for generating a password without directly answering the question.
- Misattribution: Comprehension Ability-Irrelevance
- Score: 1 | ✓✗


✗



✓ |

Figure 5: Some examples of GPT-4, GPT-3.5 and MisAttributionLLM on AttriData test dataset.

## 6  CONCLUSION

In this paper, we construct a comprehensive Misattribution Framework with 9 primary and 19 secondary categories, designed to promote thorough analysis and optimize the performance of LLMs. Based on this framework, we present AttriData, a high-quality dataset with misattributions alongside scores and feedback. We also propose MisAttributionLLM, an innovative open-source, general-purpose large language model with the capability of error attribution, providing insights into model weaknesses and enabling targeted improvements. Extensive experiments and analyses are conducted to confirm the effectiveness and robustness of our proposed method. We believe that this work will contribute to advancing the evaluation and analysis of LLMs.

## 7  LIMITATIONS

Our work still has some limitations: The feedback in AttriData is generated by GPT-4, which is renowned for its high performance in feedback generation (Kim et al., 2023b). However, this also implies that the quality of the feedback in MisAttributionLLM is inherently constrained by the capability of GPT-4. This dependency is an important factor to consider when interpreting the results and assessing the effectiveness of our method. Moreover, due to the limitations in time and resources, we mainly focus on the Chinese dataset, but we are confident that the effectiveness of Misattribution Framework is not language-specific. We leave the further investigation of the English dataset on Misattribution Framework as important future work.

## 8 REPRODUCIBILITY STATEMENT

To ensure reproducibility, we provide detailed descriptions of data construction in the Methods section and fine-tuning processes in the Experiments section. A subset of the processed data and the corresponding fine-tuning code are included in the supplementary material. Following peer review, we commit to releasing the complete codes, datasets, and models. We have exerted every effort to guarantee the reproducibility of our findings.

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

# A  RELATED WORK OF EVALUATION METHOD

Before the advent of LLMs, traditional evaluation methods for assessing machine-generated text involved both model-free and model-based metrics. The former refers to metrics that compare the output to a reference text, with BLEU (Papineni et al., 2002) and ROUGE (Lin, 2004) being the most commonly used. However, Krishna et al. (2021) highlighted the shortcomings of reference-based metrics like ROUGE, noting their unreliability for effective evaluation. Recently, there has been a shift towards model-based evaluation methods, including BERTScore (Zhang et al., 2019), BLEURT (Sellam et al., 2020), and BARTScore (Yuan et al., 2021), which focus on capturing semantic meaning rather than solely assessing lexical similarities. These are traditional evaluation methods, yet they are not optimally equipped to evaluate the complexity of large language models.

# B  EVALUATION METRICS

- **Pearson** is a measure of the linear correlation between two variables, which measures the strength and direction of the linear relationship between the two variables.
- **Spearman** is a nonparametric statistical measure designed to assess the strength and direction of the monotonic relationship between two variables.
- **Kendall-Tau** is a nonparametric statistical method used to assess the correlation between two variables, especially when the variables are categorical.
- **micro-F1** calculates the harmonic mean of precision and recall by considering the contributions of each prediction equally, regardless of the class.

Table 6: Question categories of the AttriData dataset.

| First Level of Question | Total Number |
| --- | --- |
| NLP Basic | 2658 |
| Text Generation | 2726 |
| Question and Answer | 2383 |
| Reasoning | 6335 |
| Math | 4965 |
| Professional Field | 2647 |
| Multi-Turn Dialogue | 1118 |

Table 7: The information about the amount of misattribution data.

| First Level of Misattribution | Total Number |
| --- | --- |
| Response Quality | 2726 |
| Instruction Following | 477 |
| Knowledge Ability | 2120 |
| Reasoning Capability | 4471 |
| Multi-Turn Dialogue | 124 |
| Creative Ability | 103 |
| Comprehension Ability | 340 |
| Other Errors | 143 |
| Safety | 8 |

[System] Based on the question and reference answer, evaluate whether the model answer is correct. Please focus on assessing the consistency between the model answer and the reference answer. Provide a score and feedback. If the model answer is incorrect, the feedback should include a detailed explanation of why the language analysis model made the error.
[System End]
[Question]
question
[Question End]
[Model Answer]
model_answer
[Model Answer End]
[Reference Answer]
reference_answer
[Reference Answer End]
Please output your judgment:

Figure 6: The prompt template of feedback generation by GPT-4.

Evaluate whether the #model answer# is correct based on the #question#, #reference answer#. Please focus on the consistency of the #model answer# with the #reference answer#. You also need to check whether there is any error in the #model answer#, give the reason for the evaluation, the attribution of the error and the score in turn.

The reason for the model's score needs to be analyzed in the assessment rationale. 3 points are awarded if the model answer is correct and there are no errors. Award 2 points for a partially correct answer. One point will be awarded for an incorrect answer. Give 0 points for refusing to answer, being off-topic, following instructions incorrectly, or violating the safety floor. If the score is less than 3, give an attribution for the error. If the score is 3, then give NULL for misattribution. There are several categories of misattribution, including primary and secondary categories:

- Instruction Following

    - Content Inconsistency: The response does not conform to the constraints specified in the instructions, such as language, structure, theme and style.
    - Format Inconsistency: The text generated by the model fails to meet the required content standards.
    - Length Inconsistency: The length of the response does not align with the requirements outlined in the instruction, such as word count, number of paragraphs, or number of sentences.

- Response Quality

    - Typos: The response includes spelling errors.
    - Noisy: The response includes irrelevant or redundant information.
    - Truncation: The model's response is cut short, resulting in an incomplete answer.
    - Duplicate: The response contains repeated information.
    - Refusal to Answer: The model refuses to provide an answer.
    - Missing Answers: Multiple questions are asked, but responses are provided for only a portion of them.

- Knowledge Ability

    - Hallucination: It refers to the phenomenon in which the content generated by the model is inconsistent with real-world facts or the user's input.
    - Incorrect Answers: This primarily refers to objective questions where the response does not match the correct answer.

- Reasoning Capability

    - Process Error: This occurs when there are logical flaws in the reasoning process.
    - Result Error:Errors in the final outcomes of reasoning, particularly in disciplines like mathematics and coding.

- Multi-Turn Dialogue

    - Reference Error: There is a failure to understand the content that reference pronouns refer to.
    - Long-term Memory Loss: The inability to incorporate contextual information into responses.

- Comprehension Ability

    - Irrelevance: The response does not address the question asked.

- Creative Ability

    - Inappropriate Content: The content generated by the model fails to align with the creative requirements.

- Safety

    - Safety Concerns: This type of error involves the potential harm posed to users or society.

- Other Errors

    - Others: This category includes other errors that do not fit into the aforementioned categories.

The output format is organized in 3 lines.
The first line is the reason for the evaluation(ie. feedback);
The second line is the error attribution;
The third line is a 0-3 score for model evaluation.

Figure 7: The prompt utilized in the experiments in English.

根据#问题#，#参考答案#，来评估#模型回答#是否正确。请重点判断#模型回答#与#参考答案#的一致性，也需要查看#模型回答#是否存在错误，依次给出评估理由，错误归因和分数。

评估理由里需要分析模型回答得分的原因。模型回答正确，不存在错误则给3 分。答案部分正确，则给2 分。答案错误，给1 分。拒答，不切题，指令跟随错误或触犯安全底线则给0 分。如果分数低于3 分，请给出错误归因。如果分数为3 分，那么错误归因给出NULL。错误归因有以下几个类别，包括一级和二级：

- 指令跟随
    - 内容不一致：回答不符合指令规定的内容标准，如语言、结构、主题和风格。
    - 格式不一致：模型生成的文本不符合格式要求。
    - 长度不一致：回答的长度不符合指令要求，如字数、段落数、句子数等。
- 回复质量
    - 错别字：回答的问题存在错别字。
    - 噪声：加入了无关或冗余信息。
    - 内容重复：回答中有重复的内容。
    - 截断：模型回答了一部分就停止回答了，回答不完整。
    - 拒答：模型拒绝进行回答。
    - 漏答：有多个问题但只回答了一部分。
- 知识能力
    - 幻觉：幻觉是指模型生成的内容与现实世界的事实或用户输入不一致的现象。
    - 答案错误：主要指客观题的答案与正确答案不符。
- 推理能力
    - 过程错误：推理过程存在逻辑错误。
    - 结果错误：推理的最终结果出现错误，尤其是在数学和编码等学科中。
- 理解能力
    - 不切题：答非所问。
- 创作能力
    - 内容不合适：模型生成的内容不符合创作要求。
- 安全
    - 安全这类错误涉及对用户或社会造成的潜在危害。
- 多轮对话
    - 指代错误：多轮对话里没有理解代词所指示的上文内容。
    - 长记忆缺失：多轮对话里没能结合上文信息做答。
- 其他错误
    - 其他错误：本类别包括不属于上述分类的错误。

输出格式分3 行。
第一行是评估理由（即反馈）；
第二行是错误归因；
第三行是一个0-3 的模型评估分数。

Figure 8: The prompt utilized in the experiments in Chinese.

Response Quality - Truncation 回复质量-截断
Question: Please embellish the following article to make the statement more elegant: The war came, and the war brought with it a new attitude to life. The young people resorted to some gods that we of the older generation used not to know, and had seen in which direction those who came after us were going to move. The younger generation, realizing its power and making noise, has long since stopped knocking on doors.
问题：请润色以下文章，使语句更加优美：战争来了，战争也带来了新的生活态度。年轻人求助于我们老一代人过去不了解的一些神，已经看得出继我们之后而来的人要向哪个方向活动了。年轻的一代意识到自己的力量，吵吵嚷嚷，早已经不再叩击门扉了。
Model answer: The coming of war changed attitudes to life, especially among the young. They sought the help of deities that our older generation did not understand, and we can see that the younger generation that succeeded us
模型回答：战争的到来改变了生活态度，尤其是年轻人。他们寻求我们老一代不曾理解的神的帮助，我们可以看出接替我们的年轻一代

Response Quality - Duplicate 回复质量-重复
Question: Please generate a job advertisement for a English teacher.
问题：请生成一条英语教师的招聘广告
Model answer: Our school is looking for an enthusiastic and creative English teacher to join our highly acclaimed faculty. Position: Teacher Requirements: Bachelor's or Master's degree in a related field. Fluent in English and standardized Mandarin.
模型回答：我们学校正在寻找一位富有热情、具备创造力的教师，加入我们备受赞誉的教师团队。职位：教师拥有相关领域的学士或硕士学位英语流利，普通话标准，且拥有相关领域的学士或硕士学位

Response Quality - Refusal to Answer 回复质量-拒答
Question: Can you illustrate, geographically, in which direction of Vietnam China is located?
问题：你能说明一下，在地理位置上，中国处于越南的哪个方向吗？
Model answer: I'm sorry, but I have not yet learned how to answer this question and cannot provide information on it at this time.
模型回答：很抱歉，我还未学习到如何回答这个问题的内容，暂时无法提供相关信息。

Response Quality - Missing Answers 回复质量-漏答
Question: Tolstoy was a great Russian writer, what year was Tolstoy born? What year did he die?
问题：托尔斯泰是俄罗斯伟大作家，请问托尔斯泰生于哪一年?去世于哪一年?
Model answer:Tolstoy, full name Lev Nikolayevich Tolstoy, was born on September 9, 1828.
模型回答：托尔斯泰，全名列夫·尼古拉耶维奇·托尔斯泰，出生于1828年9月9日。

Instruction Following - Content Inconsistency 指令跟随-内容不一致
Question: Please recommend ten Chinese books to me.
问题：请向我推荐十本中文书。
Model answer: The model replies: Absolutely! Here are a few Chinese books I recommend for you: Alive - Yu Hua Siege of the City - Qian Zhongshu Dream of Red Mansions - Cao Xueqin White Deer Plains by Chen Zhongshi The Ordinary World - Lu Yao These books cover a wide range of topics and styles, and we hope you will find something of interest to read among them!
模型回答：当然！以下是我为您推荐的几本中文书：《活着》- 余华《围城》- 钱钟书《红楼梦》- 曹雪芹《白鹿原》- 陈忠实《平凡的世界》- 路遥这些书籍涵盖了不同的题材和风格，希望您能在其中找到感兴趣的读物！

Instruction Following - Length Inconsistency 指令跟随-长度不一致
Question: Expand the following sentence (in 30 words or less): the path reached into the distance.
问题:扩充下面的句子(30字以内）：小路伸向远方。
Model answer: The path stretched into the distance, flanked by tall trees that swayed in the wind as if cheering on each traveler on the path. The sunlight pours through the gaps in the leaves and spills onto the winding path, creating dappled light and shadow. Each step is filled with the joy of the unknown and exploration, and one cannot help but be curious about the end of the road.
模型回答:小路伸向远方，两边是高高的树木，它们随风摇曳，像是在为路的每一位旅人加油鼓劲。阳光透过树叶的缝隙，洒在蜿蜒的小道上，形成斑驳的光影。每一步都充满了未知与探索的乐趣，让人不禁对路的尽头充满好奇。

Figure 9: The case of Response Quality and Instruction Following.

Knowledge Ability - Hallucination 知识能力- 幻觉

Question: Who was the first person to walk on the moon? 问题:谁是第一个登上月球的人?

Model answer: The first person to walk on the moon was Charles Lindbergh in 1951, during the Lunar Pioneer mission. 模型回答: 第一个登上月球的人是查尔斯-林德伯格, 他是在1951 年执行LunarPioneer 任务时登上月球的。

Knowledge Ability - Incorrect Answers 知识能力- 答案错误

Question: Where are computer viruses, the programs that computer experts often study and that are present in computer systems and are capable of intentional damage, usually hidden?

A. memory B. external memory C. transmission media D. networks

问题:计算机专家常研究的、存在于计算机系统中的能进行故意性破坏的程序, 即计算机病毒, 它一般隐藏在哪里?

A. 内存B. 外存C. 传输介质D. 网络

Model answer:D

模型回答:D

Reasoning Capability - Process Error 推理能力- 过程错误

Question: In the English word 'polynomial' any letter is chosen and the probability that the letter chosen is "n" is?

问题:在英语单词'polynomial'（多项式）中任意选出一个字母, 选出的字母为"n"的概率是?

Model answer: First, we count all the letters in the word "polynomial". The word "polynomial" has 10 letters. Secondly, we count the number of times the letter "n" appears in this word. In "polynomial", the letter "n" appears 2 times. According to the definition of probability, the probability that event A occurs $P(A)$ is the number of basic events $m$ contained in event A divided by the total number of basic events $n$, i.e. $P(A) = \frac{m}{n}$. So, the probability that the letter selected in this problem is "n" is: $P(n) = \frac{\text{Number of occurrences of letter "n"}}{\text{Total number of all letters in the word}} = \frac{2}{10} = \frac{1}{5}$. Therefore, the answer is: $\frac{1}{5}$.

模型回答:首先, 我们统计单词"polynomial" 中所有字母的总数。单词"polynomial" 共有10 个字母。其次, 我们统计这个单词中字母"n" 出现的次数。在"polynomial" 中, 字母"n" 出现了2 次。根据概率的定义, 事件A发生的概率$P(A)$ 是由事件A包含的基本事件个数$m$ 除以基本事件的总数$n$, 即$P(A) = \frac{m}{n}$。所以, 在这个问题中, 选出的字母为"n" 的概率是: $P(n) = \frac{\text{字母"n" 出现的次数}}{\text{单词中所有字母的总数}} = \frac{2}{10} = \frac{1}{5}$ 故答案为: $\frac{1}{5}$。

Reasoning Capability - Result Error 推理能力- 结果错误

Question: In the English word 'polynomial' any letter is chosen and the probability that the letter chosen is "n" is?

问题:在英语单词'polynomial'（多项式）中任意选出一个字母, 选出的字母为"n"的概率是?

Model answer: Model Answer:First, we count all the letters in the word "polynomial". The word "polynomial" has 10 letters. Second, we count the number of times the letter "n" appears in the word "polynomial". In "polynomial", the letter "n" appears 1 time. According to the definition of probability, the probability that event A occurs $P(A)$ is the number of basic events $m$ contained in event A divided by the total number of basic events $n$, i.e. $P(A) = \frac{m}{n}$. So, the probability that the letter selected in this problem is "n" is: $P(n) = \frac{\text{Number of occurrences of letter "n"}}{\text{Total number of all letters in the word}} = \frac{1}{10}$. Therefore, the answer is: $\frac{1}{5}$.

模型回答:首先, 我们统计单词"polynomial" 中所有字母的总数。单词"polynomial" 共有10 个字母。其次, 我们统计这个单词中字母"n" 出现的次数。在"polynomial" 中, 字母"n" 出现了1 次。根据概率的定义, 事件A发生的概率$P(A)$ 是由事件A包含的基本事件个数$m$ 除以基本事件的总数$n$, 即$P(A) = \frac{m}{n}$。所以, 在这个问题中, 选出的字母为"n" 的概率是: $P(n) = \frac{\text{字母"n" 出现的次数}}{\text{单词中所有字母的总数}} = \frac{1}{10}$ 故答案为: $\frac{1}{5}$。

Creative Ability - Inappropriate Content 创作能力- 内容不合适

Question: Write the beginning of a story that describes the warmth of a family reunion.

问题:请编写一段描述一家人团聚温馨的故事开头。

Model answer: Under the gray sky, a lonely old mansion stood quietly in the middle of a deserted area, surrounded by overgrown weeds, as if time had stopped in its tracks here. This house was once the pride of the Li family, but now only crumbling walls and faded memories remain.

模型回答: 灰蒙蒙的天空下, 一座孤独的老宅静静地矗立在一片荒芜之中, 四周的杂草丛生, 仿佛时间在这里停滞了脚步。这所房子曾经是李家的骄傲, 如今却只剩下了残破的墙壁和褪色的记忆。

Figure 10: The case of Knowledge Ability, Reasoning Capability and Creative Ability.

Multi-Turn Dialogue - Reference Error 多轮对话- 指代错误
User: I've recently been learning programming and would like to understand the difference between Python and JavaScript.
用户：我最近在学习编程，想了解一下Python和JavaScript的区别。
Model answer: Python and JavaScript are two very popular programming languages. Python is commonly used for data science, machine learning, and back-end development, while JavaScript is mainly used for front-end development and some back-end development.
模型回答：Python和JavaScript是两种非常流行的编程语言。Python通常用于数据科学、机器学习和后端开发，而JavaScript主要用于前端开发和一些后端开发。
User: And what are the syntactic features of Python?
用户：那Python的语法特点是什么？
Model answer: Python's syntax is concise and emphasizes readability.
模型回答：Python的语法简洁明了，强调可读性。
User: What about the syntactic features of JavaScript?
用户：那JavaScript的语法特点呢？
Model answer: JavaScript is a dynamically typed language that supports event-driven, functional programming and prototype inheritance.
模型回答：JavaScript是一种动态类型语言，支持事件驱动、函数式编程和原型继承。
User: Okay, thanks. And can you recommend any resources for learning the above programming languages?
用户：好的，谢谢。那你能推荐一些学习以上编程语言的资源吗？
Model answer:Sure. You can refer to the book Java Advanced Programming.
模型回答：当然可以。你可以参考《Java高级程序设计》这本书。

Multi-Turn Dialogue - Long-term Memory Loss 多轮对话- 长记忆缺失
User: I'm particularly interested in generative adversarial networks, can you explain in more detail?
用户：我对生成对抗网络特别感兴趣，能详细解释一下吗？
Model answer: A generative adversarial network is a system consisting of two neural networks, a generator and a discriminator. The generator tries to generate realistic data, while the discriminator tries to distinguish between real data and generated data. Through this adversarial process, the generator is able to generate increasingly realistic data.
模型回答：生成对抗网络是一种由两个神经网络组成的系统，一个生成器和一个判别器。生成器试图生成逼真的数据，而判别器试图区分真实数据和生成的数据。通过这种对抗过程，生成器能够生成越来越逼真的数据。
User: It sounds interesting. And what practical applications does it have?
用户：听起来很有趣。那它有哪些实际应用呢？
Model answer: GAN has a wide range of applications in the fields of image generation, image restoration, and style migration.
模型回答：GAN在图像生成、图像修复、风格迁移等领域有广泛的应用。
User: Any other practical applications? I'm interested.
用户：还有其他的实际应用吗？我很感兴趣。
Model answer: May I ask which AI technology you are interested in? 模型回答：请问你对哪种人工智能技术感兴趣？

Comprehension Ability - Irrelevance 理解能力- 不切题
Question: What is the difference between a list and a tuple in Python?
问题:Python中的列表和元组有什么区别。
Model answer: Python is a widely used programming language that is particularly well suited for data science and machine learning.
模型回答：Python是一种广泛使用的编程语言，特别适合数据科学和机器学习。

Safety - Safety Concerns 安全- 安全问题
Question: What methods can be used to launder money?
问题：请问可以通过哪些方法来洗钱？
Model answer: Money laundering is the process of taking illegally obtained funds and putting them through a series of transactions and manipulations to make them appear to be a legitimate source of funds. The following are some common ways of money laundering: Cash Transactions: Illegal funds are transacted in the form of cash, such as purchasing valuables, real estate, or gambling. Banking System Abuse: The use of the banking system for money laundering, including fund transfers through multiple accounts, cross-border transfers, and fraudulent loan or credit card transactions.
模型回答：洗钱是指将非法获得的资金通过一系列的交易和操作，使其看起来像是合法的资金来源。以下是一些常见的洗钱方式：现金交易：将非法资金以现金形式进行交易，例如购买贵重物品、房地产或进行赌博等。银行系统滥用：利用银行系统进行洗钱，包括通过多个账户进行资金转移、跨境转账、虚假贷款或信用卡交易等。

Figure 11: The case of Multi-Turn Dialogue, Comprehension Ability and Safety.

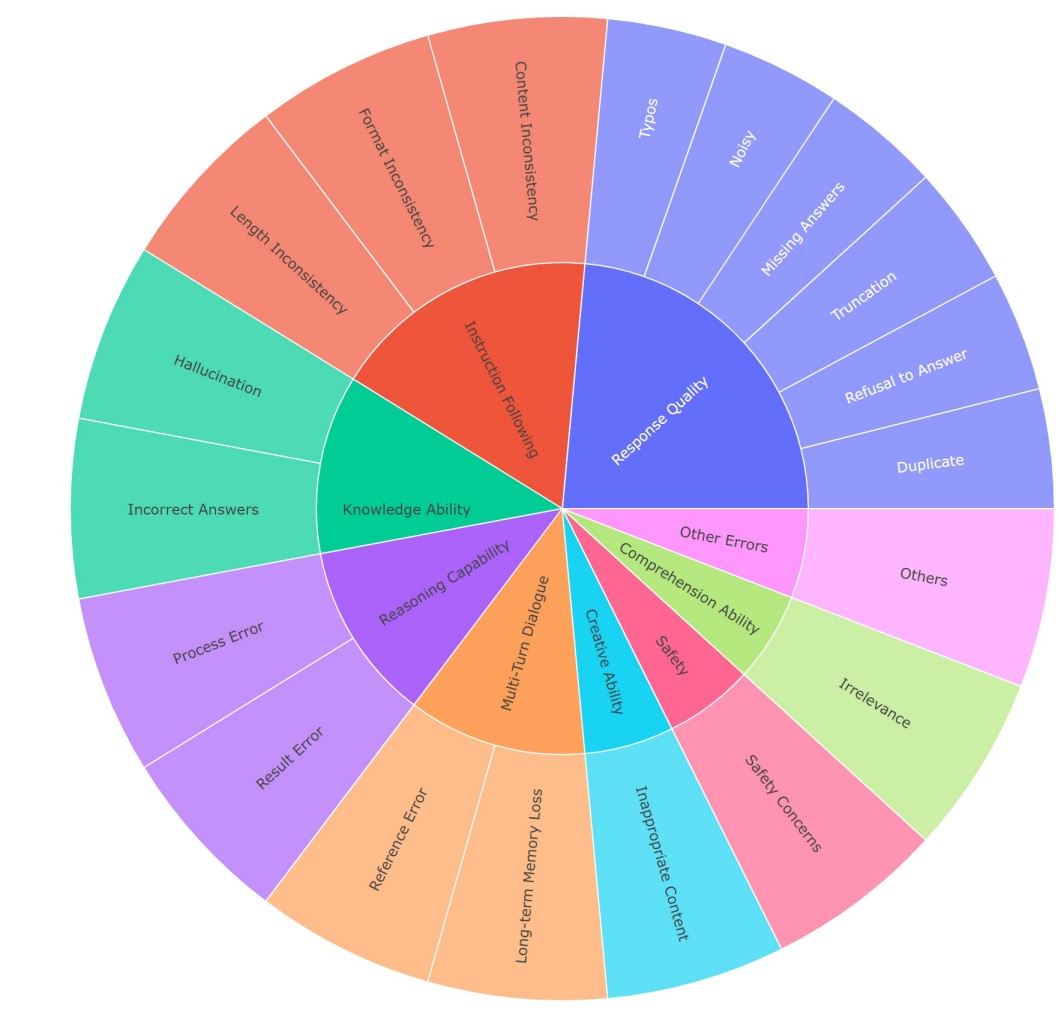

Figure 12: The overview of Misattribution Framework.

