# OpenReview forum: "MisAttributionLLM: Integrating Error Attribution Capability into LLM Evaluation"
_ICLR.cc/2025/Conference — Submitted to ICLR 2025_

### Official Review · Reviewer_3tki · 2024-10-20

**Soundness:** 2
**Presentation:** 3
**Contribution:** 3
**Rating:** 8
**Confidence:** 3

**Summary:**

The author propose a framework, a dataset and a model to deal with the problem of misattribution consists of 9 primary and 19 secondary categories. The Misattribution Framework provide fine-grained labels for the misattribution problem (instead of a score). The AttriData dataset developed for incorporating error attribution capability into LLM evaluation. The author also train model with the AttriData dataset (named MisAttributionLLM), and the experiment shows that the MisAttributionLLM trully have the ability to judge error attribution.

**Strengths:**

1. Fine-grained error attribution is a important and interesting topic.


2. MisAttributionLLM prove it can prediction error attribution in answer better then GPT-4

**Weaknesses:**

1. There is no judgment about whether GPT-4's feedback is reasonable or not in AttriData training set.

2. The instruction and question collection process is not clear. I can't figure out how the author collect them and if them may have copyright (or other) issues.

**Questions:**

1. What is the category design principle in the Misattribution Framework? Maybe different people have different opinions on the category design. (I admit that this question does not have a ground truth answer, and I won't lower the score because of this question, but I think it's interesting to discuss this question in the paper.)

2. Introduce Pearson, Spearman, and Kendall-Tau in appendix.

3. How many data have multiple error attribution in the AttriData dataset? It is also an important part of the statistics.

4. (minor) Change the example in Figure 2 to a multiple Misattribution QA example. Or it may mislead the reader to think that one question only has one error attribution.

---

> ### Author Response · Authors · 2024-11-24
> **Thank you for your valuable feedbacks.**
>
> Thank you very much for your valuable feedback. We apologize for any confusion caused by certain details in the paper. We will address each of your concerns and provide detailed explanations to help you better understand the contributions of this paper step by step:
>
> ---
>
> Q1: There is no judgment about whether GPT-4's feedback is reasonable or not in AttriData training set.
>
> A1:
> Thank you for your valuable suggestion.
>
> We selected GPT-4 for generating the feedback due to its proven efficiency and accuracy in handling generative annotation tasks, a choice supported by prior research such as Prometheus [1] and pandaLM [2]. To ensure the quality and reliability of the generated feedback, we incorporated a rigorous human review process where all feedback was meticulously examined by human annotators.
>
> [1] Kim S, Shin J, Cho Y, et al. Prometheus: Inducing fine-grained evaluation capability in language models[C]//The Twelfth International Conference on Learning Representations. 2023.
>
> [2] Wang Y, Yu Z, Yao W, et al. PandaLM: An Automatic Evaluation Benchmark for LLM Instruction Tuning Optimization[C]//The Twelfth International Conference on Learning Representations.
>
> ---
>
> Q2:
> The instruction and question collection process is not clear. I can't figure out how the author collect them and if them may have copyright (or other) issues.
>
> A2:To ensure a structured and diverse dataset, we drew inspiration from TencentLLMEval and manually labeled our data according to their task tree. This method involves categorizing each instruction or question into specific tasks or domains to maintain consistency and relevance. We took care to use original content or public domain material where possible, and for any third-party content, we ensured compliance with applicable copyright laws and obtained necessary permissions to avoid legal issues. Our goal is to respect intellectual property rights while creating a valuable resource for research and development purposes.
>
> ---
>
> Q3:
> What is the category design principle in the Misattribution Framework? Maybe different people have different opinions on the category design.
>
> A3:Thank you for your insightful feedback.
> The category design principle within the Misattribution Framework is indeed to cover as many issues as possible that can arise in the responses of LLMs. Our approach is grounded in a comprehensive analysis of extensive online user data, as well as researching relevant papers to ensure that we accurately reflect the spectrum of user queries and the nuanced errors that can arise from LLMs.
>
> The primary goal of our category design is to achieve a broad and inclusive coverage that captures the diverse range of issues users might encounter. This involves not only identifying common patterns of misattribution but also being sensitive to less frequent or more subtle types of errors. We strive to make our categories both meaningful and actionable, ensuring they can inform improvements in the model and provide valuable insights for researchers and practitioners.
>
> We acknowledge that there may be multiple valid ways to categorize these issues, and we welcome discussions and contributions from the community to refine and expand our framework. Thank you very much for your valuable feedback.
>
> ---
>
> Q4: Introduce Pearson, Spearman, and Kendall-Tau in appendix.
>
> A4:  Thank you for your valuable suggestion. We have added an introduction to the Pearson, Spearman, and Kendall-Tau indicators in Appendix B.
>
> ---
>
> Q5:
> How many data have multiple error attribution in the AttriData dataset? It is also an important part of the statistics.
>
> A5:
> Thank you for your valuable suggestion. In the AttriData dataset, a total of 2031 data points have multiple error attributions. We have included this information in Section 3.3, "DATASET ANALYSIS: Dataset Statistics."
>
> ---
>
> Q6:Change the example in Figure 2 to a multiple Misattribution QA example. Or it may mislead the reader to think that one question only has one error attribution.
>
> A6: Thank you for your valuable suggestion. We have modified Figure 2 to include a multiple misattribution QA example, ensuring that readers do not misunderstand that each question can have only one error attribution.
>
> ---
>
> Thank you again for your time and effort in reviewing our paper.

---

> > ### Comment · Reviewer_3tki · 2024-11-25
> >
> > Thanks for author's response. It do solve my concern, I have increased my score to 8.

---

> > > ### Author Response · Authors · 2024-11-25
> > > **Thank you very much**
> > >
> > > We are genuinely grateful for the time and effort you invested in reviewing our work, and are truly delighted by your strong endorsement of our research and rebuttal. Thank you very much!

---

### Official Review · Reviewer_QSCX · 2024-10-24

**Soundness:** 2
**Presentation:** 1
**Contribution:** 2
**Rating:** 5
**Confidence:** 4

**Summary:**

The paper is about improving LLM-as-a-Judge models. The authors curated a large (22K) dataset of LLM-generated outputs in Chinese (or mostly Chinese). They asked 48 human annotators to provide a general score (0-3) and detect misattributions (19 error types predefined by the authors). Then, they fine-tuned 7b LLM on this dataset and conducted experiments with 7 LLM baselines to (1) measure the correlation between the LLM and human general score and (2) measure misattribution detection. The authors show their fine-tuned LLM performs better on the test set of their dataset than other baselines.

**Strengths:**

* This dataset presents a large collection of human-annotated response quality scores, including misattribution annotations. It is an impressive and timely contribution that will undoubtedly be valuable to other researchers. The labor invested in creating this dataset is unparalleled compared to other preference-based open-source datasets.
* The experiments include numerous (7) baselines.

**Weaknesses:**

**Presentation and Missing Details:** The presentation of this paper is lacking, with numerous crucial details missing that are necessary for a proper evaluation. For example (and this is a recurring issue throughout the paper):
* What exactly are the 22K outputs evaluated by the annotators? How were these outputs generated?
* Where did the prompts of those examples originate from? Which models were used?
* What are the dataset statistics? Are all examples in Chinese? If so, what is the language distribution, and why is it only mentioned in the limitations section? This is vital information that both reviewers and readers need in order to properly interpret the results.
* What does a training example look like? How is it used for fine-tuning (this should be illustrated with an example in the appendix)?
* What exactly are the Misattribution Detection and Multi-Classification tasks?
* How are the final LLM answers extracted? What prompts were used for scoring? Are they different from those used for other models?

These are just some key aspects missing from the paper. Please refer to all the suggestions I provided to help improve the overall presentation.

**Irrelevant Misattributions and Their Statistics:** The authors do not provide any statistics on the misattributions, such as the proportion of each type or the correlations between them. I believe that some of these misattributions, like typos or duplications, may not be relevant to current LLMs. Additionally, it's unclear whether the more interesting misattributions, such as hallucinations, process errors, or issues related to long-term memory, were correctly annotated. How can I be certain that the annotators are qualified for this task? The general knowledge of current LLMs often surpasses that of the average human or crowd worker, and it’s unclear how non-expert annotators can reliably identify all hallucinations. This issue should either be clarified in the rebuttal or discussed as a limitation in the paper.

**Related Work**: All the introduction and related work citations are from 2023-2024. It gives the impression that the authors did not conduct an extensive literature review before or during their research. There is a comprehensive body of literature on error analysis in generation models that could have potentially led to a better misattribution framework (e.g., in SumEval, there are coherence and fluency dimensions that are not fully covered by the current framework). I believe the authors could have drawn inspiration from a decade of NLG research.

**Comparison with Other (OOD) Datasets:** The only notable finding from this study is yet another reinforcement of the well-known fact that fine-tuned models perform better in-domain than general LLMs (i.e., if the training and test data are the same, a fine-tuned model will outperform even a larger, more general LLM). This is a well-established result. I would like to compare MisAttributionLLM and other LLMs on different datasets, especially since dozens of preference datasets are available. Furthermore, the ablation study suggests that fine-tuning the model with misattributions does not lead to significant improvement (at best, a 0.01 improvement, which could be due to chance or because the authors likely paid more attention and ran more experiments with the full version).

**Questions:**

**Suggestions:**

Sort citations order by year (e.g.,  (Xie et al., 2023; Chang et al., 2024; Liu et al., 2024;) instead of (Chang et al., 2024; Liu et al., 2024; Xie et al., 2023)

033: I don't understand what "to guide the LLM in its continuous improvement" means.

036-038: Need rephrasing.

118: Judge LLMs that beside a score and output feedback, can identify misattributions however - it is given in natural language. The main difference is that the missattributions need to be extracted from the feedback, unlike what you propose -- but it is better to write it clearly.

120: Error attribution has been researched for decades in NLG, many papers break the evaluation into different dimensions.  Scoring low in the dimension can be thought of as an error.
For example:

https://arxiv.org/abs/2307.10928

https://arxiv.org/abs/2310.00752

https://aclanthology.org/2022.acl-long.531/

I would emphasize that several misattributions can occur at the same time.

257: Kappa is on a -1 to 1 scale and is not percentages.

266: Nothing is "illustrated"  in Figure (2).

267-270: The letter notations are not used, so I think you can omit it.

Caption of Figure 2: Please provide more details in the caption.

320: What is the fine-tuning data? what does it contain? How a does data instance look? Can you provide examples in the appendix? I can't understand what is contains... all the components from the inference? Only the misattributions?

What about dataset statistics? The distribution of the misattribution? Correlation between misattribution? What are the correlations between each misattribution and the score?

Table 2: Remove "_" from column names.  What exactly does the "Human(GPT-4 assisted)" mean? Is it only because GPT-4 generates feedback, or do annotators use GPT-4 in their annotations? Please answer here and use the caption to elaborate, this is important (also for GPT-4(human assistant, I can't understand what it means...)

What are the prompts you use for the LLMs? Do you use few-shot? Any advanced LLM-as-a-Judge technique?

What is the language of the MisAttri dataset? How many English examples there are? How many Chinese? Any other languages? Can you provide statistics?

351: I guess every 2nd-year student knows what it F1 is; you can remove the five sentences explaining it to save space.

348: What is the difference between Misattribution Detection and Multi-Classification? Why do you write five sentences about the F1 score but not elaborate on the specific prediction tasks in your paper? I guess that for Misattribution Detection, the task of the LLM is to predict True or False for each error type (do you use all the 19 error types in the prompt? one by one? Do you ask the LLM to explain? Many missing information...). I guess you specify the error types in the prompt for the Multi-Classification and ask it to mention only relevant errors. Again, I can't evaluate the paper correctly if I can't understand what you did, the tasks, and how exactly you have implemented the experiments (I don't care about the hardware and number of epochs,... I am asking for much more basic information; how can I tell that you don't have flaws in the instruction of the prompt or how you evaluate and extract LLM answers?).


Figure 3: Notice that there is no left or right, but upper and lower model.

Figure 4: I do not like the "radar charts", I (personally) cannot read them - I suggest using barplots for each misattribution category (thus supporting comparison between models).

---

> ### Author Response · Authors · 2024-11-24
> **Thank you for your valuable feedbacks.**
>
> Thank you very much for your valuable feedback. We apologize for any confusion caused by certain details in the paper. We will address each of your concerns and provide detailed explanations to help you better understand the contributions of this paper step by step:
>
> ---
>
> Q1:
> Presentation and Missing Details: The presentation of this paper is lacking, with numerous crucial details missing that are necessary for a proper evaluation. For example (and this is a recurring issue throughout the paper):
>
> Thank you for your valuable suggestion.
>
> ---
>
> q1:What exactly are the 22K outputs evaluated by the annotators? How were these outputs generated?
>
> a1:The outputs are feedback, misattribution and score. The process of generation we illustrated in the section Annotation Workflow. Error attribution and scores are manually labeled and feedback is generated by GPT-4. In our process, human annotations haved reviewed all the feedback.
>
> ---
>
> q2:Where did the prompts of those examples originate from? Which models were used?
>
> a2:The prompts used in the examples provided in Figures 6.
> We used ERNIE Bot2[1] and Hunyuan[2] models.
>
> [1]https://yiyan.baidu.com
>
> [2]https://hunyuan.tencent.com
>
> ---
>
> q3:What are the dataset statistics? Are all examples in Chinese? If so, what is the language distribution, and why is it only mentioned in the limitations section? This is vital information that both reviewers and readers need in order to properly interpret the results.
>
> a3:The dataset statistics, including details on subject categories and misattribution, are provided in Tables 6 and 7. The majority of the dataset is in Chinese, with the language distribution being predominantly Chinese. Specifically, there are 1,321 entries in English, which we have included in the data statistics.
>
> ---
>
> q4:What does a training example look like? How is it used for fine-tuning (this should be illustrated with an example in the appendix)?
>
> a4:For a training sample, given a instruction, a question, a model answer text, and a reference text, the objective is to produce a comprehensive result including a rating score, a misattribution, and feedback (In the paper “3.4 FINE-TUNING LANGUAGE MODEL”).
> For illustration, consider the following example:
>
> ●Instruction: Refer to Figure 7 in the paper for the detailed context(too long).
>
> ●Question: Who was the first person to walk on the moon?
>
> ●Model Answer: The first person to walk on the moon was Charles Lindbergh in 1951, during the LunarPioneer mission.
>
> ●Reference Text: Neil Armstrong was the first person to walk on the moon in 1969 during the Apollo 11 mission.
>
> The output includes:
>
> ●Feedback: The model answered incorrectly. Neil Armstrong was the first person to walk on the moon, suggesting a potential hallucination problem.
>
> ●Misattribution: Knowledge Ability-Hallucination
>
> ●Score: 1
>
> We provide a visual representation of this process in Figure 2 of the paper. Additionally, more detailed examples are available in the Appendix, specifically in Figures 9, 10, and 11.
>
> ---
>
> q5：What exactly are the Misattribution Detection and Multi-Classification tasks?
>
> a5：For misattribution detection, this refers to whether the judge model correctly determines that there is an error in the model response. For multi-classification of misattribution, this refers to the process of error attribution (i.e. whether the error is correctly categorized). We added the task introduction in our paper in line 366-368 ( section The Performance of Error Attribution).
>
> ---
>
> q6: How are the final LLM answers extracted? What prompts were used for scoring? Are they different from those used for other models?
>
> a6: The final LLM answers are extracted along with the feedback, misattribution, and score, which are provided simultaneously. The prompts used for scoring are detailed in Figures 7 and 8. All modeling experiments use a uniform input format, ensuring consistency across different models.

---

> ### Author Response · Authors · 2024-11-24
> **Response (2)**
>
> Q2:Irrelevant Misattributions and Their Statistics: The authors do not provide any statistics on the misattributions, such as the proportion of each type or the correlations between them. I believe that some of these misattributions, like typos or duplications, may not be relevant to current LLMs. Additionally, it's unclear whether the more interesting misattributions, such as hallucinations, process errors, or issues related to long-term memory, were correctly annotated. How can I be certain that the annotators are qualified for this task? The general knowledge of current LLMs often surpasses that of the average human or crowd worker, and it’s unclear how non-expert annotators can reliably identify all hallucinations. This issue should either be clarified in the rebuttal or discussed as a limitation in the paper.
>
> A2:
> Thank you for your valuable suggestion.
> We clarify that we have provided information about dataset statistics, including details on subject categories and misattribution in Tables 6 and 7.
>
> The primary goal of our error category design is to achieve broad and inclusive coverage. We have observed that various error types often co-occur; for example, a model response may simultaneously exhibit issues with response quality and hallucination.  We aim to highlight that each type of error has its own focus.
>
> Regarding the qualifications of the annotators, we have implemented a rigorous training and review process to ensure high-quality annotations. This includes multiple rounds of training sessions and multi-turn reviews to maintain consistency and accuracy. The detailed methodology is outlined in the "Annotation Workflow" section of our paper. We believe this process significantly enhances the reliability of our annotations, even when dealing with sophisticated phenomena such as hallucinations.
> We hope these clarifications help alleviate your concerns.
>
> ---
>
> Q3:Related Work: All the introduction and related work citations are from 2023-2024. It gives the impression that the authors did not conduct an extensive literature review before or during their research. There is a comprehensive body of literature on error analysis in generation models that could have potentially led to a better misattribution framework (e.g., in SumEval, there are coherence and fluency dimensions that are not fully covered by the current framework). I believe the authors could have drawn inspiration from a decade of NLG research.
>
> A3: Thank you for your detailed review and valuable comments on our paper.
> I would like to clarify that not all of the work we have cited is from 2023-2024; there are other more recent works in Appendix A of related work.
> We have chosen to cite literature from 2023-2024 primarily to ensure that our research reflects the most recent research advances, especially those findings that are directly related to our research questions. However, we also realize that this approach may lead readers to believe that we are ignoring the relevant theoretical foundations that have been accumulated over time.
>
> We carefully read the relevant paper you mentioned, which provides some special perspectives for the evaluation. We have included the following paper in the related work section.
> We believe that these additions will enhance both the depth and breadth of our research, presenting a more comprehensive picture of the existing analysis of errors in generative models.

---

> ### Author Response · Authors · 2024-11-24
> **Response (3)**
>
> Q4:Comparison with Other (OOD) Datasets: The only notable finding from this study is yet another reinforcement of the well-known fact that fine-tuned models perform better in-domain than general LLMs (i.e., if the training and test data are the same, a fine-tuned model will outperform even a larger, more general LLM). This is a well-established result. I would like to compare MisAttributionLLM and other LLMs on different datasets, especially since dozens of preference datasets are available. Furthermore, the ablation study suggests that fine-tuning the model with misattributions does not lead to significant improvement (at best, a 0.01 improvement, which could be due to chance or because the authors likely paid more attention and ran more experiments with the full version).
>
> A4: Thank you for your valuable suggestion.
> First, we want to clarify that our study was not aimed at comparing the capabilities of LLMs, but rather at assessing whether the model's outputs can be correlated with human scoring. The reason we did not conduct such a comparison is that other preference datasets lack error attribution, which is a critical component of our research.
> We agree that your point is reasonable, and therefore, we conducted experiments using the AlignBench dataset.
> Performance on Alignbench:
> | LLMs    | Pearson | Spearman | Kendall-Tau |
> |---------|---------|----------|-------------|
> | Qwen2-7B   | 0.179   | 0.154    | 0.136       |
> | GPT-3.5  | 0.333   | 0.337    | 0.306       |
> | GPT-4    | 0.756   | 0.761    | 0.747       |
> | MisattributionLLM | 0.739 | 0.743 | 0.729 |
>
> It can be found that our method is comparable to GPT-4 in terms of correlation with human scoring.
>
> Regarding the ablation experiment, since misattribution is only part of the output. It can be interpreted as having a supporting role. Misattribution and scores are not strongly correlated.
> Additionally, we did not use the full version, which is a basic experimental  literacy.
>
> ---
>
> Q5:
> 033: I don't understand what "to guide the LLM in its continuous improvement" means.
>
> A5: We apologize for the unclear writing. Obtaining accurate error attribution helps us to quickly pinpoint the problem. Subsequently, we can improve the model's ability by choosing whether the improvement is needed in pre-training, SFT (Supervised Fine-Tuning), or post-training. With this systematic approach to error attribution, we are able to more accurately guide model iteration and optimization.
>
> ---
>
> Q6:120: Error attribution has been researched for decades in NLG, many papers break the evaluation into different dimensions. Scoring low in the dimension can be thought of as an error. For example:
> https://arxiv.org/abs/2307.10928
> https://arxiv.org/abs/2310.00752
> https://aclanthology.org/2022.acl-long.531/
> I would emphasize that several misattributions can occur at the same time.
>
> A6:
> Thank you for your valuable suggestion.  For a detailed response, please refer to our earlier answer in A3 for the same answer.
>
> ---
>
> Q7:320: What is the fine-tuning data? what does it contain? How a does data instance look? Can you provide examples in the appendix? I can't understand what is contains... all the components from the inference? Only the misattributions?
>
> A7:  Please refer to the response in A1-a4 and A1-a6 above for  answer.
>
> ---
>
> Q8:What about dataset statistics? The distribution of the misattribution? Correlation between misattribution? What are the correlations between each misattribution and the score?
>
> A8:Please refer to the response in A2 above for  answer.
>
> Correlation Between Misattribution and Score: We have already outlined the relationship between error attribution and the score in the Annotation Workflow section. Specifically, if the score is less than 3 points, annotators are required to identify the types of errors in the model's answer using the Misattribution Framework.
>
> ---
>
> Q9:Table 2: Remove "_" from column names. What exactly does the "Human(GPT-4 assisted)" mean? Is it only because GPT-4 generates feedback, or do annotators use GPT-4 in their annotations? Please answer here and use the caption to elaborate, this is important (also for GPT-4(human assistant, I can't understand what it means...)
>
> A9: Thank you for your valuable suggestion. We have removed the underscores "_" from the column names in Table 2.  Yes, it is because feedback was generated with GPT-4.
> GPT-4 assisted indicates that the feedback was generated with the help of GPT-4, whereas human assisted means that humans were involved in reviewing the scoring.
> We hope this clarification helps.
>
> ---
>
> Q10:What are the prompts you use for the LLMs? Do you use few-shot? Any advanced LLM-as-a-Judge technique?
>
> A10:  We have added the prompts used for the LLMs in Figures 7 and 8. We did not use few-shot. We employed the Chain of Thought.

---

> ### Author Response · Authors · 2024-11-24
> **Response (4)**
>
> Q11:What is the language of the MisAttri dataset? How many English examples there are? How many Chinese? Any other languages? Can you provide statistics?
>
> A11: Please refer to the response in  A1-a3 above for the same answer.
>
> ---
>
> Q12:348: What is the difference between Misattribution Detection and Multi-Classification? Why do you write five sentences about the F1 score but not elaborate on the specific prediction tasks in your paper? I guess that for Misattribution Detection, the task of the LLM is to predict True or False for each error type (do you use all the 19 error types in the prompt? one by one? Do you ask the LLM to explain? Many missing information...). I guess you specify the error types in the prompt for the Multi-Classification and ask it to mention only relevant errors. Again, I can't evaluate the paper correctly if I can't understand what you did, the tasks, and how exactly you have implemented the experiments (I don't care about the hardware and number of epochs,... I am asking for much more basic information; how can I tell that you don't have flaws in the instruction of the prompt or how you evaluate and extract LLM answers?).
>
> A12: Thank you for your valuable suggestion.
> We understand that the lack of detailed explanation regarding detection and multi-classification may have caused some confusion, and we sincerely apologize for this oversight. To clarify, in our paper, we introduce the detection and multi-classification task in lines 366-368 of the 'The Performance of Error Attribution' section.
>
> As you accurately guessed, our approach involves providing the large language model (LLM) with a prompt containing all 19 error types. The LLM then evaluates each data, providing feedback, misattribution identification, and a score all at once.
>
> To enhance the transparency and reproducibility of our work, we have included the exact prompts utilized for the LLMs in Figures 7 and 8 of the manuscript. Additionally, Section 3.4, 'Fine-Tuning Language Model', now contains more comprehensive details about our experimental setup and procedures.
>
> We hope these clarifications provide a clearer understanding of our methodology and help in evaluating our research more effectively. If you have any further questions or require additional information, please do not hesitate to contact us.
>
> ---
>
> Thank you for your valuable suggestion. We have refined the writing that you mentioned:
> - Sort citations order by year (e.g., (Xie et al., 2023; Chang et al., 2024; Liu et al., 2024;) instead of (Chang et al., 2024; Liu et al., 2024; Xie et al., 2023)
> - 036-038: Need rephrasing.“As most of the top-performing LLMs, such as GPT-4 (Achiam et al., 2023), can only be accessed via the OpenAI API because its closed-source.”
> - Kappa is on a -1 to 1 scale and is not percentages.
> - 266: Nothing is "illustrated" in Figure (2).
> - 267-270: The letter notations are not used, so I think you can omit it.
> - Caption of Figure 2: Please provide more details in the caption.
> - 351: I guess every 2nd-year student knows what it F1 is; you can remove the five sentences explaining it to save space.
> - Figure 3: Notice that there is no left or right, but upper and lower model.
> - Figure 4: I do not like the "radar charts", I (personally) cannot read them - I suggest using barplots for each misattribution category (thus supporting comparison between models).
>
> ---
>
> Thank you again for your time and effort in reviewing our paper.

---

> > ### Comment · Reviewer_QSCX · 2024-11-25
> > **Thanks**
> >
> > Thank you for the response. I believe the paper has improved; however, I still think it could be strengthened with additional work. That said, I will raise my score to reflect my appreciation for your efforts in addressing the feedback.

---

> > > ### Author Response · Authors · 2024-11-25
> > > **Thank you very much**
> > >
> > > Thank you for your thoughtful response and for raising your score. We are genuinely grateful for the time and effort you invested in reviewing our work. We value your constructive feedback and are pleased to hear that you appreciate the improvements made. We understand your perspective that the paper could be further strengthened with additional work, and we are committed to making the necessary revisions to ensure it meets the highest standards of excellence. Thank you very much!

---

### Official Review · Reviewer_wEpB · 2024-11-04

**Soundness:** 2
**Presentation:** 3
**Contribution:** 2
**Rating:** 5
**Confidence:** 4

**Summary:**

This paper proposes a framework which consists of 9 primary and 19 secondary categories to facilitate the evaluation of LLMs. Using this framework, it then introduces a dataset for incorporating error attribution capability into LLM evaluation. Finally, it fine-tunes Qwen2-7B with the proposed dataset to obtain MisAttributionLLM for fine-grained evaluation. MisAttributionLLM  shows a strong correlation with human evaluators.

**Strengths:**

1. By fine-tuning on the proposed AttriData, the authors are able to reach better performance than baselines on most of the metrics, which demonstrate the utility of AttriData. I like the fact that the authors fine-tune an LLM based on the proposed dataset.

2. Based on Section 3, the dataset construction process seems to be rigorous, so dataset quality is not my concern.

**Weaknesses:**

1. The impact of this paper is limited. As stated in the limitation section, the paper’s primary focus is on a Chinese-language dataset, which restricts the generalizability of the findings. Each language presents unique linguistic features. For example, even when all the error types of your framework exist in English, these error types might appear as different forms than the examples you see in Chinese. Thus, your findings might vary in a different language.

2. This seems to be an incremental work with low novelty. It is not clear to me the difference between the dataset of "Evaluating llms at detecting errors in llm responses" and your dataset.

3. Although the authors claim a comprehensive Misattribution Framework, this framework still appears coarse. I do not see concrete examples of each error type and some of the error types (e.g., hallucination) might need to be further expanded. Please see my question below.

4. Certain error categories, such as those involving multi-turn dialogue and instruction following, may be underrepresented in the proposed dataset, which limits the usability of AttriData. As discussed in line 463, insufficient training data for these two aspects might be a problem for fine-tuning LLMs.

**Questions:**

1. Would you please clarify line 123? For the paper "Evaluating llms at detecting errors in llm responses", you mentioned that "the types of error they identify are limited". So what is difference of error types between your framework/dataset and theirs? Is this difference really important? I think their dataset uses English so a wider range of audience can be benefited.

2. Since you only work on Chinese-language dataset, I suggest to restrict your findings to Chinese and mention this important information at the very beginning of your paper (e.g., title, abstract, and/or introduction) to make your writing clear. I suggest not to claim something that you have not done.

3. Hallucination is just one error type in your framework. What if the content generated by the model is inconsistent with real-world facts but is still consistent with the user’s input? I suggest to dive deeper in some types such as hallucination error. There are two papers I have read before about taxonomy of hallucination that might help: "FaMeSumm: Investigating and Improving Faithfulness of Medical Summarization" and "Understanding Factuality in Abstractive Summarization with FRANK: A Benchmark for Factuality Metrics".

4. What are the "realistic evaluation scenarios" mentioned in line 211 and 212?

---

> ### Author Response · Authors · 2024-11-24
> **Thank you for your valuable feedbacks.**
>
> Thank you very much for your valuable feedback. We apologize for any confusion caused by certain details in the paper. We will address each of your concerns and provide detailed explanations to help you better understand the contributions of this paper step by step:
>
> ---
>
> Q1:The impact of this paper is limited. As stated in the limitation section, the paper’s primary focus is on a Chinese-language dataset, which restricts the generalizability of the findings. Each language presents unique linguistic features. For example, even when all the error types of your framework exist in English, these error types might appear as different forms than the examples you see in Chinese. Thus, your findings might vary in a different language.
>
> A1:Thank you for your valuable suggestion.
> Our study primarily focuses on Chinese language datasets, which we acknowledge may limit the general applicability of our findings to some extent. However, we believe that while each language has its unique characteristics, there are also significant commonalities across languages. Many types of errors observed in Chinese responses are similarly present in English and other languages. Therefore, **we emphasize that the Misattribution Framework is designed to be universally applicable.**
>
> We also mention in the limitations section that we plan to apply our framework to other languages and compare error types and processing strategies across languages. We believe that through this cross-language comparison, we can better understand the impact of language-specific factors on error analysis and further optimize our framework.
>
> We believe that this study provides valuable insights into Chinese language error attribution and lays the foundation for future cross-linguistic research.
>
> ---
>
> Q2:This seems to be an incremental work with low novelty. It is not clear to me the difference between the dataset of "Evaluating llms at detecting errors in llm responses" and your dataset.
>
> A2:
> Errors of detection and errors of attribution are two different levels of problems, with errors of attribution being a further deepening and refinement of errors of detection.
>
> Our dataset is based on the Misattribution Framework, the first open-source, general-purpose large language model capable of error attribution. It is specifically designed for fine-grained evaluation. **The framework provides a wider range of error types and a deeper analysis of language model behavior for error attribution.**
>
> ---
>
> Q3:Although the authors claim a comprehensive Misattribution Framework, this framework still appears coarse. I do not see concrete examples of each error type and some of the error types (e.g., hallucination) might need to be further expanded. Please see my question below.
>
> A3:
> Thank you for your valuable suggestion.
> We recognize that the description of the Misattribution Framework in the paper may appear cursory. To address this issue, **we have added specific examples and detailed descriptions of each error type in Appendix Figure 9, 10, and 11.** We believe these additions will help readers better understand our framework.
>
> ---
>
> Q4:Certain error categories, such as those involving multi-turn dialogue and instruction following, may be underrepresented in the proposed dataset, which limits the usability of AttriData. As discussed in line 463, insufficient training data for these two aspects might be a problem for fine-tuning LLMs.
>
> A4:
> Thank you for your valuable suggestion.
> In the AttriData dataset, error categories related to multi-turn dialogue and instruction following might be underrepresented. Given the relative complexity of these tasks, we plan to expand the dataset in future releases by adding more data related to multi-turn dialogue and instruction following. This will include a wider range of scenarios and more complex dialog structures to enhance the comprehensiveness and usability of the dataset.

---

> ### Author Response · Authors · 2024-11-24
> **Response (2)**
>
> Q5:Would you please clarify line 123? For the paper "Evaluating llms at detecting errors in llm responses", you mentioned that "the types of error they identify are limited". So what is difference of error types between your framework/dataset and theirs? Is this difference really important? I think their dataset uses English so a wider range of audience can be benefited.
>
> A5：In the paper "Evaluating LLMs at Detecting Errors in LLM Responses," the authors define four types of errors and use a detection method to determine whether an error has been made. This approach often involves checking each error multiple times. In contrast, our framework and dataset include a richer and more comprehensive set of error types, allowing us to recognize multiple errors simultaneously.
>
> We believe this difference is significant because it enables a more thorough evaluation and understanding of the errors that LLMs can generate in real-world applications. By covering a wider range of error attributions, we can better assess the robustness and reliability of LLMs. Additionally, by identifying and categorizing these errors, we can provide valuable guidance for the future design and improvement of LLMs, helping developers build more robust and accurate models.
>
> Regarding the language concern, please refer to the response in A1 above for a detailed explanation.
>
> ---
>
> Q6:Since you only work on Chinese-language dataset, I suggest to restrict your findings to Chinese and mention this important information at the very beginning of your paper (e.g., title, abstract, and/or introduction) to make your writing clear. I suggest not to claim something that you have not done.
>
> A6:
> Thank you for your valuable suggestion. We added the Chinese language restriction.
>
> ---
>
> Q7:Hallucination is just one error type in your framework. What if the content generated by the model is inconsistent with real-world facts but is still consistent with the user’s input? I suggest to dive deeper in some types such as hallucination error. There are two papers I have read before about taxonomy of hallucination that might help: "FaMeSumm: Investigating and Improving Faithfulness of Medical Summarization" and "Understanding Factuality in Abstractive Summarization with FRANK: A Benchmark for Factuality Metrics".
>
> A7:Thank you for your valuable suggestion.
> We generally categorize issues like the one you mentioned, where the model's output is inconsistent with real-world facts but consistent with the user’s input, as incorrect answers to knowledge competency questions. Our initial goal was to create a comprehensive framework that was not exhaustively categorized in some details. We recognize that detailed exploration of specific error types, such as hallucination, requires further attention. We plan to refine and expand our framework in future research to address these nuances.
>
> Thank you for the references you provided. Papers such as "FaMeSumm: Investigating and Improving Faithfulness of Medical Summarization" and "Understanding Factuality in Abstractive Summarization with FRANK: A Benchmark for Factuality Metrics" have been very helpful. Based on your suggestions and our ongoing literature review, we will delve deeper into hallucination errors in our future research.
>
> ---
>
> Q8:What are the "realistic evaluation scenarios" mentioned in line 211 and 212?
>
> A8: The "realistic evaluation scenarios" correspond to the statement in line 158: "User-Driven Focus: The data focuses on issues that are of significant concern to users. Originating from real-world application scenarios, it provides an accurate reflection of the current public demand for LLMs."
> We have removed the detailed description from our data statistics to improve clarity and representation.
>
> ---
>
> Thank you again for your time and effort in reviewing our paper.

---

> ### Author Response · Authors · 2024-11-30
> **Looking Forward to Your Response**
>
> Dear Reviewer wEpB,
>
> We sincerely appreciate your time and effort in reviewing our paper and offering valuable suggestions.
> As the author-reviewer discussion phase is drawing to a close, we would like to confirm whether our responses have effectively addressed your concerns.
>
> We provided detailed responses to your concerns a few days ago, and we hope they have adequately addressed your issues. If you require further clarification or have any additional concerns, please do not hesitate to contact us. We are more than willing to continue our communication with you. Thank you very much for your time and consideration. I am looking forward to hearing from you.
>
> Best regards

---

> > ### Comment · Reviewer_wEpB · 2024-12-02
> >
> > Thanks the authors for the rebuttal and reminder! I will improve the presentation score. I think the detailed exploration of specific error types and cross-language comparison are very important, as you aim to propose a comprehensive framework. Both breadth and depth are of high importance.

---

> > > ### Author Response · Authors · 2024-12-03
> > > **Thanks**
> > >
> > > Thank you for your valuable feedback. We are committed to ensuring that our framework maintains both breadth and depth, and we will continue to enhance our presentation to reflect this comprehensively. If any remaining issues require further clarification or improvement, we would be grateful if you could point them out. We are committed to making all necessary refinements. Thanks.

---

### Official Review · Reviewer_8pkz · 2024-11-04

**Soundness:** 3
**Presentation:** 2
**Contribution:** 2
**Rating:** 5
**Confidence:** 4

**Summary:**

In this paper, the authors first propose a taxonomy of LLM errors with 9 primary and 19 secondary categories, then annotate a collection of LLM error responses with these categories and use GPT-4 to further write feedback. Finally, the authors fine-tune a 7B model on the annotated dataset and achieve a higher attribute capability than other baselines without finetuning.

**Strengths:**

1. The idea of implementing large-scale error attribution is relatively novel.

2. The dataset is relatively large with manually annotated data points.

**Weaknesses:**

1. The research question and motivation is a bit unclear to me. In the introduction, although the authors write ''This oversight tends to result in a misjudgment of the LLM’s performance and is likely to hinder the identification of critical opportunities for improvement.'', it is still unclear what is the importance and significance of the research question. What will happen if we only detect the errors without identifying 19 categories?

2. The biggest concern is the dataset creation. Although the authors collect a wide range of datasets, the rule-based classification that classifies all tasks of the LLMs into 7 categories is not comprehensive enough. For instance, what about the code completion task? For math, there are many types of questions, such as math word problems, GSM8k, etc. It is unclear how the authors collect the datasets and how to cover all tasks with these categories. Even if the dataset is perfect, the categorization is also subjective without good reasoning and classification methods. This makes it a bit difficult to judge the following results of the proposed work.

3. Some experiments are unreasonable. First, naturally, the performance of fine-tuned models is better than zero-shot settings. Can you try to compare it with the few-shot setting of GPT-4 to get the result? Also, do you feed the instructions of each type to GPT-4? Otherwise, it is unfair for other models to compare with.  Furthermore, the accuracy of Misattribution Detection is close to 100, I am wondering if the setting and categories are too simple for LLMs. Finally, multi-classification has not been introduced before, making it difficult to understand the results.

4. The GPT-4 based feedback is doubtful. There is no human annotation to ensure the quality of the feedback. At least some sample and annotated work is needed. Besides, do you train the model on feedback? If so, is the model better than GPT-4 generated feedback?

**Questions:**

See the weakness.

---

> ### Author Response · Authors · 2024-11-24
> **Thank you for your valuable feedbacks.**
>
> Thank you very much for your valuable feedback. We apologize for any confusion caused by certain details in the paper. We will address each of your concerns and provide detailed explanations to help you better understand the contributions of this paper step by step:
>
> ---
>
> Q1:The research question and motivation is a bit unclear to me. In the introduction, although the authors write ''This oversight tends to result in a misjudgment of the LLM’s performance and is likely to hinder the identification of critical opportunities for improvement.'', it is still unclear what is the importance and significance of the research question. What will happen if we only detect the errors without identifying 19 categories?
>
> A1:
> Thank you for your valuable suggestion.
> Our research question and motivation come from the fact that due to online platforms (e.g., Hunyuan[1], Doubao[2], and Wenxin[3]) a huge amount of user questions and model answers are generated every day. **In order to efficiently analyze the performance of models and pinpoint problems in their capabilities**, we need to develop an automated error classification system to systematically categorize and attribute errors. By analyzing the ‘bad cases’—cases where the model fails to handle inputs correctly—we can more accurately diagnose performance issues.
>
> If we rely on manual annotation to identify and classify these errors, the results are often **difficult to standardize and categorize due to subjectivity and varying criteria, which can greatly reduce efficiency and accuracy.**
> Obtaining accurate error attribution helps us to quickly pinpoint the problem. Subsequently, we can improve the model's ability by choosing whether the improvement is **needed in pre-training, SFT (Supervised Fine-Tuning), or post-training.**
> With this systematic approach to error analysis and attribution, we are able to more accurately guide model iteration and optimization, thereby accelerating the model improvement process and increasing its overall performance and user satisfaction.
>
> If we only detect errors without identifying all 19 error categories, we will be unclear about the shortcomings of the large model's ability to respond and may miss out on a full understanding of LLM performance. Each error category points to a different weakness in the model, and ignoring these categories can lead us to fail to identify potential opportunities for model improvement in specific domains or tasks. For example, if we fail to identify errors related to reasoning ability, we may not be able to target improvements in the model's reasoning ability, which is critical in many real-world applications.
>
> [1]https://hunyuan.tencent.com
>
> [2]https://doubao.com
>
> [3]https://yiyan.baidu.com
>
> ---
>
> Q2:The biggest concern is the dataset creation. Although the authors collect a wide range of datasets, the rule-based classification that classifies all tasks of the LLMs into 7 categories is not comprehensive enough. For instance, what about the code completion task? For math, there are many types of questions, such as math word problems, GSM8k, etc. It is unclear how the authors collect the datasets and how to cover all tasks with these categories. Even if the dataset is perfect, the categorization is also subjective without good reasoning and classification methods. This makes it a bit difficult to judge the following results of the proposed work.
>
> A2:
> Thank you for your valuable suggestion.
> This categorization, which we have explained in the paper, is inspired by TencentLLMEval and AlignBench to cover the task as comprehensively as possible. We manually labeled the data based on TencentLLMEval's task tree.  We have added data sources in section dataset construction.
>
> We categorize code completion tasks under professional competencies. This includes subcategories such as code generation, defect location, programming basics, and code completion tasks. We also include math questions from elementary, middle, and high school, covering geometry, algebra, and probability. These questions include multiple-choice and quiz formats.
>
> Classification Methods: After extensive research on benchmarks for large model tasks and in-depth discussions with several senior data scientists, we have finalized our classification methods to ensure a rich and comprehensive categorization.

---

> ### Author Response · Authors · 2024-11-24
> **Response (2)**
>
> Q3:Some experiments are unreasonable. First, naturally, the performance of fine-tuned models is better than zero-shot settings. Can you try to compare it with the few-shot setting of GPT-4 to get the result? Also, do you feed the instructions of each type to GPT-4? Otherwise, it is unfair for other models to compare with. Furthermore, the accuracy of Misattribution Detection is close to 100, I am wondering if the setting and categories are too simple for LLMs. Finally, multi-classification has not been introduced before, making it difficult to understand the results.
>
> A3:
> Thank you for your valuable suggestion.
> We provided each type of instruction to GPT-4 **to ensure a fair comparison with other models.** We provided each type of instruction to all models, which helps them better understand and perform specific tasks, thus ensuring a more accurate performance comparison benchmark.
>
> Detecting errors is relatively simple, requiring only binary judgments. However, attributing errors is not easy, which is the focus of our study.
>
> You mentioned that multi-classification has not been introduced before, which can lead to difficulties in understanding the results. We recognize this as an oversight and apologize for any confusion.
> The multi-classification refers to the process of error attribution (i.e. whether the error is correctly categorized).
> To clarify, in our paper, we introduce multi-classification task in lines 366-368 of the 'The Performance of Error Attribution' section.
>
> ---
>
> Q4:The GPT-4 based feedback is doubtful. There is no human annotation to ensure the quality of the feedback. At least some sample and annotated work is needed. Besides, do you train the model on feedback? If so, is the model better than GPT-4 generated feedback?
>
> A4:
> Thank you for your insightful feedback.
> We selected GPT-4 for generating the feedback due to its proven efficiency and accuracy in handling generative annotation tasks, a choice supported by prior research such as Prometheus [1] and pandaLM [2]. **To ensure the quality and reliability of the generated feedback, we incorporated a rigorous human review process where all feedback was meticulously examined by human annotators.**
>
> Furthermore, we have indeed trained our model using the feedback data. The experimental results, detailed in Figure 3, demonstrate that our model's performance is comparable to that of GPT-4. This indicates that our model can produce feedback of similar quality, thereby enhancing the overall effectiveness of our system.
>
> [1] Kim S, Shin J, Cho Y, et al. Prometheus: Inducing fine-grained evaluation capability in language models[C]//The Twelfth International Conference on Learning Representations. 2023.
>
> [2] Wang Y, Yu Z, Yao W, et al. PandaLM: An Automatic Evaluation Benchmark for LLM Instruction Tuning Optimization[C]//The Twelfth International Conference on Learning Representations.
>
> ---
>
> Thank you again for your time and effort in reviewing our paper.

---

> ### Author Response · Authors · 2024-11-30
> **Looking Forward to Your Response**
>
> Dear Reviewer 8pkz,
>
> We sincerely appreciate your time and effort in reviewing our paper and offering valuable suggestions.
> As the author-reviewer discussion phase is drawing to a close, we would like to confirm whether our responses have effectively addressed your concerns.
>
> We provided detailed responses to your concerns a few days ago, and we hope they have adequately addressed your issues. If you require further clarification or have any additional concerns, please do not hesitate to contact us. We are more than willing to continue our communication with you. Thank you very much for your time and consideration. I am looking forward to hearing from you.
>
> Best regards

---

> ### Comment · Reviewer_8pkz · 2024-12-02
> **Reviewer Response**
>
> Thanks for your detailed explanation. The authors addressed my concerns about motivation and data creation. I will raise the soundness score of the work. Thanks.

---

> > ### Author Response · Authors · 2024-12-03
> > **Thanks**
> >
> > We are pleased to hear that our revisions have successfully addressed your  concerns about motivation and data creation. If any remaining issues require further clarification or improvement, we would be grateful if you could point them out. We are committed to making all necessary refinements. Thanks.

---

### Meta-Review · Area_Chair_35Qi · 2024-12-24

**Metareview:**

This paper presents a taxonomy of different kinds of errors made by language models, encapsulates it in a dataset, and finetunes a language model to make these predictions for better evaluation of language models. Overall this work will improve the evaluation of LLM-as-a-judge approaches through extensive data collected solely for this purpose.

**Strengths:** There is a lot of value for this line of work, given the challenges of LLM evaluation, and a dataset like AttriData with human labels can be useful for further research.

**Weaknesses:** The major concern that reviewers raised is the somewhat subjective categorization of errors made by the language models and their comprehensiveness. There might be an overreliance on GPT-4 or similar language models to evaluate LLMs, which may result in a reinforcing behavior. Moreover, the paper’s primary focus on a Chinese-language dataset raised questions about the generalizability of the findings to other languages.

**Reason for rejection**: Please see the weaknesses above. With the exception of one, the reviewers were not satisfied with the depth of the work. The only positive reviewer seemed to be okay with the dataset contribution.

**Additional Comments On Reviewer Discussion:**

Based on the reviewer feedback, the authors have provided changes in the paper that clarify the motivation of the work (attribution not error labeling). Reviewers pointed out many missing details in the presentation of the work, some of which were incorporated into the revised manuscript by the authors. While the reviewers recognized the value of the dataset, the premise of the taxonomy underlying the dataset is something that they found quite arbitrary and could not support in its current form.

---

### Decision · Program_Chairs · 2025-01-22

Reject